# Folk arts-inspired twice-coagulated configuration-editable tough aerogels enabled by transformable gel precursors

Lishan Li [1], Guandu Yang[1,2], Jing Lyu[1], Zhizhi Sheng [1], Fengguo Ma[2] & Xuetong Zhang [1,3] ✉

Aerogels, as famous lightweight and porous nanomaterials, have attracted considerable attention in various emerging fields in recent decades, however, both low density and weak mechanical performance make their configuration-editing capability challenging. Inspired by folk arts, herein we establish a highly efficient twice-coagulated (TC) strategy to fabricate configuration-editable tough aerogels enabled by transformable gel precursors. As a proof of concept, aramid nanofibers (ANFs) and polyvinyl alcohol (PVA) are selected as the main components of aerogel, among which PVA forms a flexible configuration-editing gel network in the first coagulation process, and ANF forms a configuration-locking gel network in the second coagulation process. TC strategy guarantees the resulting aerogels with both high toughness and feasible configuration editing capability individually or simultaneously. Altogether, the resulting tough aerogels with special configuration through soft to hard modulation provide great opportunities to break through the performance limits of the aerogels and expand application areas of aerogels.

Nature is a world full of beauty and mystery, which contains many amazing geometric configurations. These configurations are not only informative but also of profound scientific significance. Configuration editing was accompanied by the development of the whole human civilization, such as stone tools grinding, bronze ware and pottery construction, cotton and hemp fabric weaving. In materials science, configuration editing plays an important role in structural design and functional realization of products. For example, Metamaterials can be customized with extraordinary electromagnetic, acoustic, mechanical, and thermal properties with appropriate configuration design[1–4]; Shape memory materials (SMMs) including polymers, ceramics, alloys, etc., can be restored from a programmed configuration to the initial one under stimuli, showing great potentials in flexible electronics, soft robotics, 4D printing and so on[5–7]; Liquid crystal elastomers (LCEs) are famous as a class of stimuli-responsive polymers with large and reversible configuration change via their slightly cross-linked polymer

networks, playing a critical role in robotics, microfluidics, optics, and energy fields[8–11]. On the other hand, aerogels, as the well-known porous solid-state nanomaterials with extremely low density, ultralow thermal conductivity, high specific surface area, and strong adsorption capacity, have been developed into an ideal material family applied in a variety of emerging fields including thermal insulation, energy storage, catalysis, sensor, environmental remediation[12–22]. Aerogels with simple configurations such as aerogel powder, aerogel felt, aerogel fiber, aerogel film, and aerogel monolith have been largely investigated, but the aerogels with complex configurations for breaking through performance limits and expanding application areas are urgently required.

Folk arts have played an important role in the development of science. It is not only an effective means of materials fabrication but also inspires scientific and technological innovation[23–27]. Folk arts, such as origami, weaving, and clay molding, can transform simple

[1]Suzhou Institute of Nano-Tech and Nano-Bionics, Chinese Academy of Sciences, Suzhou, PR China. [2]Key Laboratory of Rubber-Plastics (Ministry of Education), School of Polymer Science and Engineering, Qingdao University of Science and Technology, Qingdao, PR China. [3]Division of Surgery & Interventional Science, University College London, London, UK. ✉e-mail: xtzhang2013@sinano.ac.cn

configuration (thread, paper, or sheet) into three-dimensional complex ones through a multi-step editing process. Inspired by those folk arts, it should be a feasible approach to fabricate aerogels with complex configurations by configuration editing like origami, weaving, and clay molding. Nevertheless, the direct configuration editing of aerogels was undoubtedly challenging due to their low densities, weak mechanical performances, and low mechanical fatigue resistance, which couldn't undergo repeated editing processing, such as folding, stretching, compression, distortion, etc. Therefore, finding an indirect way to prepare aerogel with complex configuration is the key to solving the above problem. Likewise, when turning our attention back to folk arts, we notice that clay molding, for example, involves two stages of shaping and solidifying: first molding soft clay into various shapes and then solidifying them into hard porcelains. Inspired by those folk arts, we propose a twice coagulation (TC) strategy to edit the configurations of aerogels indirectly in gel precursors by establishing a soft elastic gel network for flexible configuration editing in the first coagulation step, and then a rigid gel network for configuration locking in the second coagulation step. To achieve this goal, PVA, as the well-known active component of elastic hydrogels, was chosen as an ideal candidate to build flexible networks for configuration editing[28–30]; ANFs, as a representative of high-strength nanofibers and structural reinforcement elements, can be used to build rigid hard networks for configuration locking[31–41].

Herein, the proposed twice coagulation (TC) strategy was successfully established to fabricate configuration-editable ANF-PVA (AP) tough aerogels enabled by transformable gel precursors (Fig. 1). To programmatically control the stepwise gelling of ANF and PVA, the ANF-PVA dispersion in dimethyl sulfoxide (DMSO) was firstly subjected to freeze-thaw cycles to obtain the flexible configuration-editable PVA organogel network containing unfixed ANFs in the first coagulation step. Then, organogel was immersed in protic solvents to promote the protonation of ANF and second network hardening, forming a configuration-locked rigid hydrogel network in the second coagulation step. After solvent exchange and supercritical drying in sequence, the cellular AP aerogel with enhanced mechanical properties was obtained with a specific tensile modulus as high as $666\,MPa\,cm^3\,g^{-1}$ and toughness of the aerogel as high as $2093\,kJ\,m^{-3}$, much superior to the current polymeric aerogels. Twice coagulation (TC) strategy, endow

the aerogels with not only mechanical enhancement but also configuration editing capability of aerogels, by which tough aerogels with various configurations such as bowl, boat, Chinese knot spring, coil, etc could be obtained. For example, the aerogel with spring configuration can deform and spring back freely such as stretching, compression, bending, twisting, etc, against the rigidity of tough aerogel without configuration editing. The tensile strain of the aerogel with coil configuration obtained by TC strategy can reach up to 7000%, whereas that of an linear aerogels by TC strategy is just about 25%. Finally, take advantage of the porosity, high mechanical strength, and unusual performance of tough aerogels with special configurations, applications of aerogel can be expanded to new fields, such as self-supporting thermal-insulation structures, thermal management devices, stimuli-responsive shape memory devices, etc. These results demonstrate a robust strategy to design tough aerogels with complex configurations for breaking through performance limits and expanding application areas.

## Results

### Twice-coagulated (TC) strategy

The specific twice-coagulated process and the network transition mechanism were primarily investigated. First of all, to determine the optimal experimental conditions, a comprehensive investigation and detailed discussion on the influence of ANF-PVA ratio, freezing to thawing cycles, and ambient conditions on the TC process were conducted and presented in the Supplementary Information (details in Supplementary Note 1, Supplementary Figs. 1–5). The optimal organogel-hydrogel-aerogel transition was performed as follows: firstly, 4.0 wt%, 4.5 wt%, 5.0 wt%, 5.5 wt%, 6.0 wt% ANF-PVA mixed solutions in alkali (KOH or t-BuOK) DMSO with 5:1 mass ratio of PVA to ANF were prepared. In the first coagulation setp, the mixed solutions were coagulated into red ANF-PVA composite organogels (AP organogels) after three freeze-thawing cycles, which need to be sealed to avoid humidity absorption from ambience. Then, in the secondary coagulation step, organogels were immersed into deionized water to produce white ANF-PVA composite hydrogels (AP hydrogels). Finally, after solvent exchange in ethanol and supercritical $CO_2$ drying in sequence, ANF-PVA composite aerogels were obtained, named AP-17 $(170\,mg\,cm^{-3})$, AP-19 $(190\,mg\,cm^{-3})$, AP-23 $(230\,mg\,cm^{-3})$, AP-25

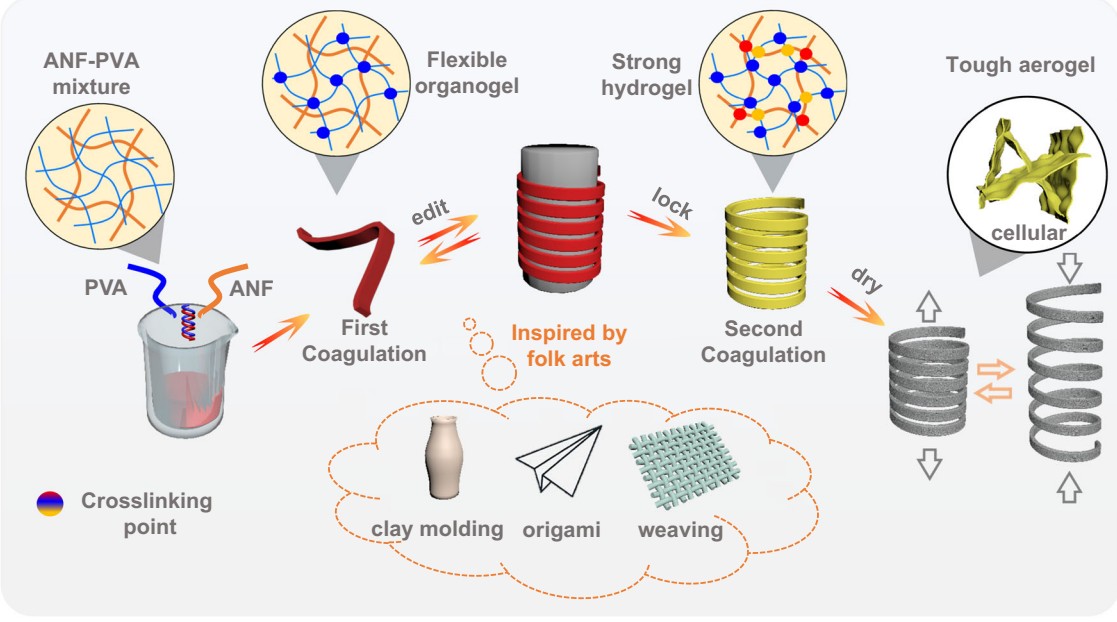

**Fig. 1 | Design and overview diagram.** Schematic illustration of folk arts-inspired, twice-coagulated, configuration-editable tough aerogels enabled by transformable gel precursors.

(250 mg cm⁻³), AP-27 (270 mg cm⁻³) aerogels in turn based on the densities (Fig. 2a, Supplementary Fig. 6).

In order to clarify the roles of PVA and ANF in the formation of the organogel in the first coagulation step, two sets of comparative experiments of PVA and ANF in KOH DMSO solution were carried out respectively, named PVA-KOH-DMSO and ANF-KOH-DMSO (Supplementary Fig. 7). After one freeze-thaw cycle, PVA-KOH-DMSO formed organogel, whereas ANF-KOH-DMSO was still in solution. This difference indicates that the forming network of red AP organogel was mainly contributed by PVA, while the ANF played little or no role, which can be further confirmed by Raman scattering spectra and UV–vis spectra. As shown in Fig. 2b, PVA has no visible scattering signal among 800–1800 cm⁻¹. The Raman spectrum of ANF-PVA mixture was similar to aramid nanofibers deprotonated by KOH reported elsewhere[42], indicating the successful deprotonation of ANF. The Raman spectrum of AP organogel remained unchanged compared with that of ANF-PVA mixture, suggesting that ANF was still in a

deprotonated state and made no contribution to the first coagulation, ensuring the construction of first PVA elastic network. To enhance the intuitiveness of our findings, color changes and UV–vis absorption spectra were employed as supporting evidence for the protonation/deprotonation state of ANF. Typically, in its deprotonated state, the ANF solution exhibits a red color and displays a prominent absorption peak at 335 nm in the UV–vis spectrum. Upon reprotonation, the solution's color transitions to a pale yellow while experiencing a significant decrease in absorption intensity. Comparative analysis of PVA-ANF mixed solutions revealed that even after undergoing freezing and thawing processes, the organic gel retained its red coloration and exhibited a similar UV absorption spectrum featuring a strong peak at 335 nm (Fig. 2c). These observations further support that ANF remained in a deprotonated state during the first coagulation step without participating in organogel network formation, consistent with Raman spectroscopy results. By the way, slight fading of color and reduction in UV absorption may be attributed to partial contact

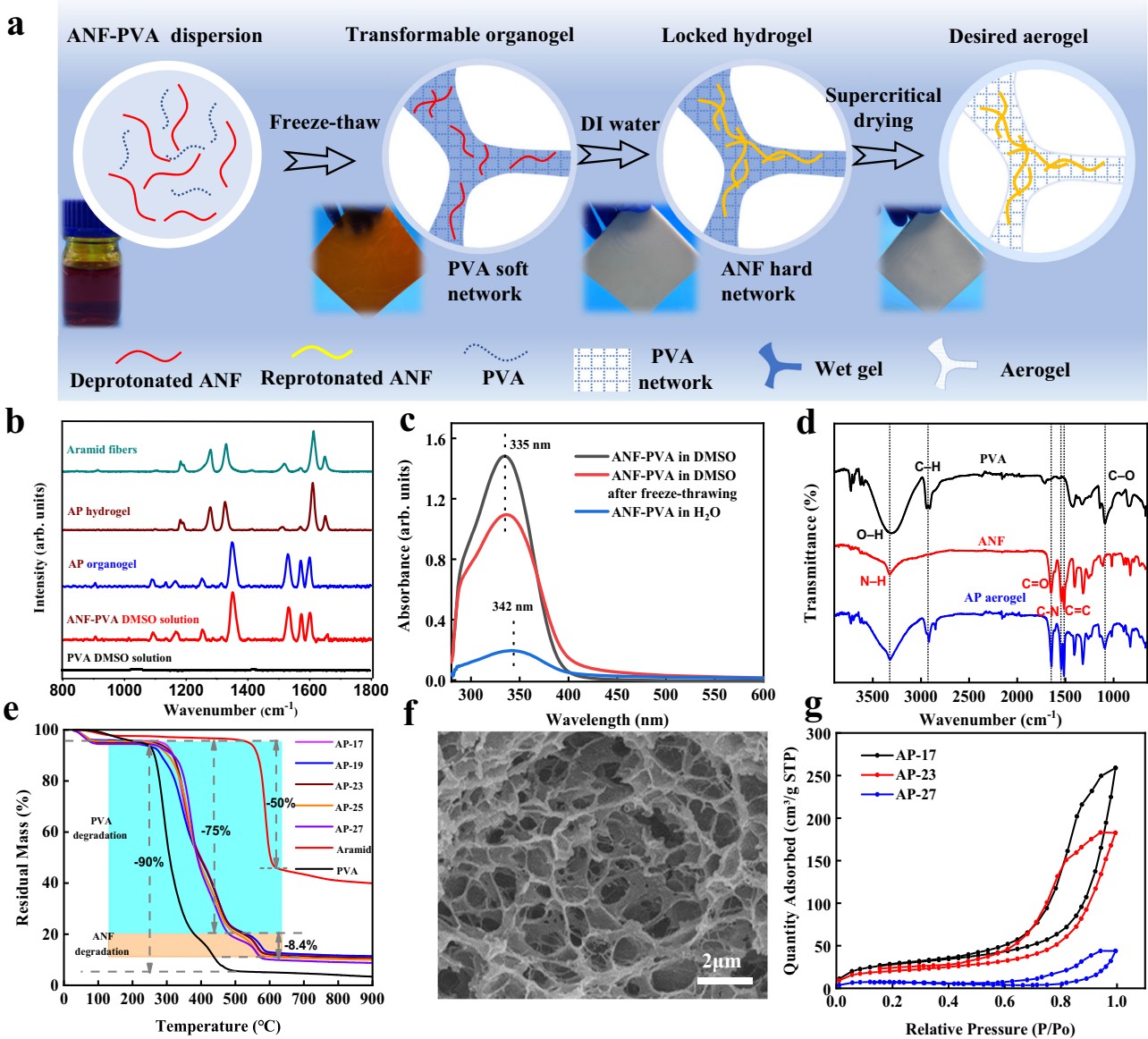

**Fig. 2 | Twice coagulated process and network transformation mechanism of the fabricated aerogels. a** Schematic diagram of the evolution of the ANF-PVA double gel-network during twice coagulation process. **b** Raman scattering of PVA DMSO solution, ANF-PVA DMSO solution, AP organogel, AP hydrogel and Aramid fibers, respectively. **c** UV-Vis spectra of diluted ANF-PVA solution, of which after freeze-thawing and after protonation in water, respectively. **d** Infrared spectra of PVA, ANF and AP aerogel, respectively. **e** TGA curves of Aramid fiber, PVA and AP-17, AP-19, AP-23, AP-25, AP-27 aerogels. **f** SEM images of AP-23 aerogel sample. **g** Nitrogen adsorption-desorption isotherm of the AP-17, AP-23, AP-27 aerogel samples.

between ANFs repelled by DMSO crystals. So, the AP organogels could be structured as PVA organogel network containing un-fixed ANFs after the first coagulation step.

To investigate the internal network of hydrogel during secondary coagulation, a series of tests and characterizations were conducted. After solvent exchange in water, the transition of AP organogel into hydrogel induced a significant shift in the Raman spectrum within the range of 800–1800 cm$^{-1}$. This observed Raman spectrum exhibited striking similarities to that of aramid fibers, providing evidence for the protonation of ANF during the second coagulation process. (Fig. 2b). Simultaneously, the color of the AP organogel changed from red to white and the UV absorption peak showed a substantial decrease (Fig. 2c), which further indicates the re-protonation process of ANF in this process. Reprotonation of ANFs leads to the elimination of repulsion between ANFs, resulting in subsequent ANF-ANF interactions facilitated by π–π interactions between aromatic rings and hydrogen bonding among amide groups. Additionally, similar to other reported studies on the regulation of intermolecular interactions or crystallization of PVA chains induced by solvent exchange[43], the solvent exchange in water also facilitates further enhancement of PVA-PVA chain interactions within the hydrogel. Compared with the weak diffraction near 20° in organogel, attributing to the crystallization in aramid nanofibers or PVA macromolecules, the stronger XRD diffraction and new forming peak near 40° indicate the further crystallization of PVA chains in hydrogel (Supplementary Fig. 8). What's more, the solvent exchange in water not only induced the reprotonation of ANFs, the crystallization of PVA, but also influence the formation of ANF-PVA hydrogen bonding. In ANF-PVA mixture and AP organogel, DMSO, as a strong hydrogen bond acceptor, inhibits hydrogen bond formation between PVA and ANF. Upon solvent exchange with water, a weaker hydrogen bond acceptor, the hydrogen-bond interaction generate between the hydroxyl groups on PVA and carbonyl groups on ANFs due to the energetically favorable OH···O=C interaction and the cooperativity of supramolecular interactions[44]. Infrared absorption spectra provide insights into the ANF–PVA interactions. It could be seen that the C=O stretching vibration peak at 1651 cm$^{-1}$ moves to 1644 cm$^{-1}$ in infrared spectrum (Supplementary Fig. 9), indicating the existence of hydrogen bonds between ANF-PVA. Therefore, to summarize, the results revealed that in the process of secondary coagulation, multiple interactions among PVA-PVA, ANF-ANF, and ANF-PVA coexist and synergistically promote the formation of a dense crosslinked network as illustrated in Supplementary Fig. 10.

The chemical nature and physical structure of obtained aerogels after drying were examined with the Fourier transform infrared spectroscopy (FTIR), thermogravimetric analysis (TGA), scanning electron microscopy (SEM) and the Brunauer-Emmett-Teller (BET) method. The infrared characteristic absorption peaks, corresponding to the stretching vibration modes of ANF (N–H 3326 cm$^{-1}$, C=O 1647 cm$^{-1}$, C–N 1541 cm$^{-1}$ and C=C 1513 cm$^{-1}$) and PVA (O–H 3303 cm$^{-1}$, C–H 2945 cm$^{-1}$ and C–O 1088 cm$^{-1}$) can be found in the FTIR spectra of AP aerogel (Fig. 2d). Meanwhile, TGA revealed a 75% weight loss of the aerogel from 250 °C to 500 °C with the thermal degradation of PVA, and 8.4% weight loss from 520 °C to 620 °C related to the thermal degradation of ANF (Fig. 2e). Theoretical mass ratio of PVA to ANF calculated with weight loss value is (75%/90%)/(8.4%/50%) ≈5/1, which is consistent with the 5:1 mass ratio of PVA to ANF in initial mixed solution (details in Supplementary equation 1). Altogether, these results confirmed the chemical composition of final AP aerogels and suggest that PVA and ANF maintained the chemical structure stability during the preparation of aerogels. From cross-section SEM image, all the aerogels AP-17, AP-19, AP-23, AP-25 and AP-27 undergoing three freeze-thaw cycles show continuous porous cellular network skeleton composed of ANFs wrapped in PVA, with a wall thickness of about 40–250 nm and a macropore size of about 500 nm to 2 μm. (Fig. 2f, Supplementary Fig. 11), which is distinct from the reported AP

nanofibrous network[45]. The generation of cellular porous structure may derive from the growth of ice crystal in freezing process, which would push aside the PVA and ANF around it. Subsequently, the ice-pushed network was fixed along with the formation of PVA and ANF network during twice coagulation. The nitrogen adsorption/desorption isotherms of the AP-17, AP-23 and AP-27 aerogels show hysteresis loops at high relative pressure, indicating the existence of plentiful mesopores (Fig. 2g). For example, the average mesoporous pore size of AP-23 aerogel is calculated to be 16 nm, with the pore size distribution ranging from 2 nm to 26 nm. Therefore, aerogels fabricated by TC strategy consist of cellular network involving continuous wall-like skeleton and hierarchical pore structure.

## Mechanical properties of AP organogel, hydrogel and aerogel

Thanks to the TC strategy, the evolution from organogel, hydrogel to aerogel will inevitably lead to incremental mechanics, which would be investigated meticulously in the following discussion. The stress-strain response of AP-23 organogel under cyclic compression-relaxation and tension-relaxation tests was firstly characterized. The overlapping cyclic compression curves from five repeated tests with 50% strain (compression modulus ≈0.015 MPa) suggested high compression elasticity of AP-23 organogel (Fig. 3a). Similarly, constant tensile stress (≈0.13 MPa) at 50% strain from fifty repeated tensile test demonstrates high tensile resilience of AP-23 organogel (Fig. 3b). These results reflect the high flexibility of organogels, which provides great probability for repeated configuration editing. Then, in order to study the mechanical differences, compressive and tensile properties in different gel states are compared. From compressive stress–strain curve in Fig. 3c, the compressive strength of AP-23 organogel, AP-23 hydrogel and AP-23 aerogel increases successively with the compressive moduli of 0.014 MPa, 1.5 MPa and 30. 9 MPa, and the stresses at 10% strain are 0.0015 MPa, 0.15 MPa and 1.3 MPa respectively. It is worth noting that the tensile strength of AP-23 organogel, hydrogel, and aerogel follows a similar increasing trend, while the elongation at break decreases. As depicted in Fig. 3d, the tensile moduli of AP-23 organogel, hydrogel, and aerogel increase from 0.3 MPa and 2.1 MPa to 87 MPa respectively; likewise, the breaking strength increases from 0.45 MPa and 1.1 MPa to 5.3 MPa accordingly; however, the elongation at break decreases from 110% and 60% to only 25%. This phenomenon can be attributed to variations in the crosslinking density. As mentioned above, organogels consist of flexible PVA networks with elastic PVA molecular chains and a low crosslinking density, enabling them to stretch extensively. The formation of a hybrid network between ANFs and PVAs in hydrogels leads to increased crosslinking densities, which subsequently reduces the elastic PVA molecular chain length between crosslinkers, thus reducing the elongation at break of the AP hydrogel according to rubber-like elasticity theory[46,47]. Similarly, after drying in supercritical CO$_2$, the swelling and amorphous PVA molecular chains in wet gels undergo further crystallization and aggregation, resulting in an additional increase in cross-linking density and restricted movement of PVA molecular chains within aerogels. Consequently, aerogels exhibit the highest mechanical strength but the lowest elongation at break among these three gel states. These conclude that the evolution from organogel, hydrogel to aerogel leads to the mechanical enhancement of the gel networks significantly.

Surprisingly, despite of the high porosity, AP aerogel has the strongest mechanical strength among these gels, which is even dozens of times higher than that of hydrogel. Motivated by this phenomenon, compressive and tensile properties of aerogels with different densities were studied in detail. All the compressive curves in Fig. 3e exhibit similar characteristics with three regions: a distinct linear beginning region, a softening middle stage and ultimately hardening stage. The distinct linear region implies that the high-strength elastic modulus of AP aerogels can withstand a large compressive load before buckling of aerogel skeleton. The compressive moduli of AP aerogels increase with

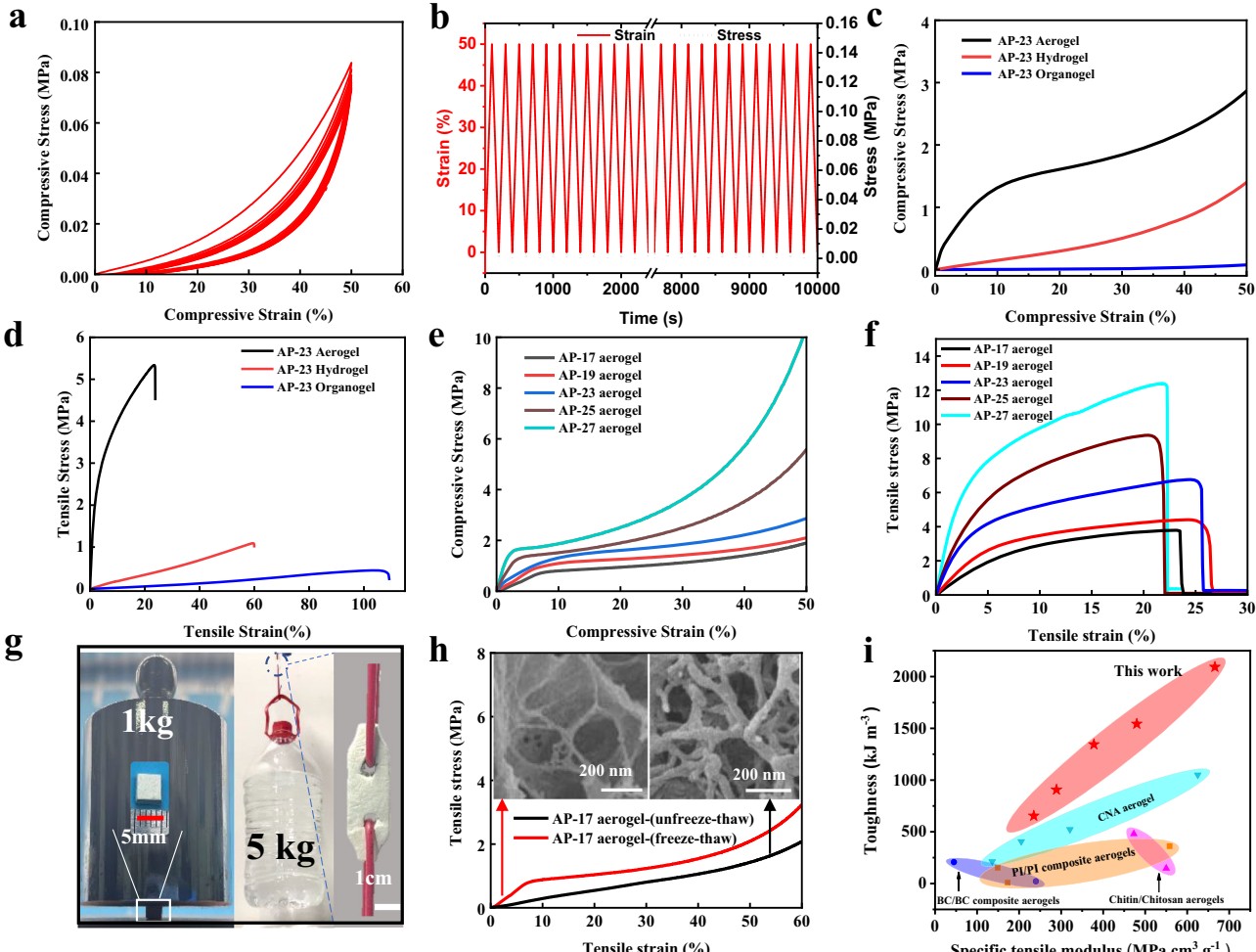

**Fig. 3 | The mechanical properties of AP organogel, hydrogel and aerogel.**
**a** Compressive cycle curve of AP-23 organogel. **b** Tensile cycle curve of AP-23
organogel. **c** Compressive stress-strain curves of AP-23 organogel, hydrogel, and
aerogels. **d** Tensile stress-strain curves of AP-23 organogel, hydrogel, and aerogels.
**e** Compressive stress-strain curves of AP-17, AP-19, AP-23, AP-25, AP-27 aerogels.
**f** Tensile stress-strain curves of AP-17, AP-19, AP-23, AP-25, AP-27 aerogels.
**g** Photographs of AP-23 aerogels under compression and tension: AP-23 aerogel
with sizes of 0.5 × 0.5 × 0.5 cm³ bearing 1 kg loading; AP-23 aerogel plate lifting 5 kg

water. **h** Compressive stress-strain curves and SEM images (insert) of AP aerogels
with and without freeze-thawing process. **i** Toughness and specific tensile modulus
of AP aerogels with different densities, as compared with other high-strength
polymeric aerogels including composite nanofiber aerogels (CNA), bacterial cel-
lulose (BC) aerogels, polyimide(PI) aerogels, and chitin/chitosan aerogels. The
toughness was calculated by integrating the tensile stress-strain curves from var-
ious materials (Supplementary Table 1).

the increase of densities, and reach a modulus as high as 86.4 MPa for
AP-27. The tensile properties of AP aerogels display a similar trend
(Fig. 3f). These aerogels exhibit increasing tensile moduli and tough-
ness with densities increasing from 170 mg cm⁻³ (AP-17) to 270 mg cm⁻³
(AP-27). Like most aerogels, log–log plots of compressive and tensile
modulus of AP vs density are linear[48]. The tensile moduli of APs as a
function of densities follow a relation E ~ ρ³⁺⁰·²⁷, (E, Young's modulus; ρ,
density) (Supplementary Fig. 12), which is a typical relation for cellular
foam materials with a random network[49].

Notably, the specific tensile modulus and toughness of AP-27
aerogel reach as high as 666 MPa cm³ g⁻¹ and 2093 kJ m⁻³, which is
equivalent to 71% of the human knee ligament toughness
(~2940 kJ m⁻³)[50]. To be more intuitive, direct reflection of high stiffness
and strength of AP aerogel was further conducted by loading experi-
ments under both compression and tension (Fig. 3g). The samples can
bear 35,000 times and lift 50,000 times mechanical loads heavier than
their own weights without severe distortion even if the aerogel is per-
forated in the middle when lifting 5 kg water. Furthermore, the AP
aerogels by TC strategy exhibit stronger mechanical properties than
general ANF-PVA (AP-G) composite aerogels and other polymeric

aerogels. The compression modulus of the AP aerogels by TC strategy is
3.6 times higher than that of the AP-G aerogels reported elsewhere
without freezing and thawing (Fig. 3h)[45]. When compared to the high-
strength polymeric aerogels reported so far (Supplementary Table 1), AP
aerogel has remarkable strength with the modulus of AP-27 as high as
180 MPa (Supplementary Fig. 13). Furthermore, AP aerogels exhibit a
dual nature of both high specific tensile modulus and toughness, among
which AP-27 aerogel was considered as highest-strength polymeric
aerogel up to now (Fig. 3i).

Subsequently, the mechanism of strength enhancement from
organogel, hydrogel to aerogel was analyzed in depth. Predictably, the
good compression resilience, tensile resilience and fatigue resistance
of AP organogel should be derived from the elastic PVA network
generated in the first coagulation as analyzed above. Then, the
mechanical enhancement from organogel to hydrogel was probably
due to the dense cross-linked network generated in the second coa-
gulation illustrated above. In contrast, the mechanical enhancement
of AP aerogels is more complex and ambiguous, which needs more focus
here. The first rational explanation for mechanical enhancement of AP
aerogel comes to our mind is the further aggregation of gel network

and crystallization of PVA during the drying process (Supplementary Fig. 14). However, more importantly, we noticed that in Fig. 3h, the freeze-thaw operation not only induces the formation of organogel networks, but also further improves the mechanical properties of aerogels. This means that the mechanical enhancement from hydrogel to aerogel is not only related to the drying process, but also to the freeze-thawing process by TC strategy.

The internal microscopic network structures of the comparative aerogels AP-FT-n ($n$ = 0, 1, 3) with n freeze-thaw cycles were further observed (Supplementary Fig. 15). From SEM images, AP-FT-0 aerogel without freeze-thaw cycles consists of fibrillar 3D skeletons, in which ANFs were wrapped by PVA at the joint. The AP-FT-1 with one freeze-thaw cycle consists of both fibrillar 3D skeletons and cellular 3D skeletons. While, the AP-FT-3 aerogel, as discussed in Fig. 2f, consists of cellular 3D skeleton, in which amounts of ANFs were wrapped by PVA wall (Fig. 3h insets, Supplementary Fig. 15). The freeze-thaw induced fiber-to-wall skeleton evolution may be attributed to repeated DMSO crystal repulsion and cross-linking of PVA during multiple freeze-thaw process. Specifically, in the absence of freeze-thaw treatment, the initially evenly dispersed ANF nanofibers and PVA in the solution was transformed into AP-FT-0 aerogel with a uniform 3D fiber network. After freezing treatment, the ANFs and PVAs were squeezed around the DMSO crystals to form a wall, meanwhile part of the PVAs were cross-linked, thereby preserving the configuration of the wall. The remaining uncrosslinked PVA/ANF walls were re-dispersed into the solution during the thawing process. Consequently, we observed the coexistence of the 3D fiber network and the wall in AP-FT-0 aerogel. As the freeze-thaw cycles increases, the DMSO crystals continue to push the ANF-PVA around, leading to generation of new walls. This resulted in fewer fibrillar 3D skeletons but an increasing presence of cellular 3D skeletons as evidenced by AP-FT-3 samples. In this process, the growth and repulsion of DMSO crystals caused aggregation of nanofibers and PVA, resulting in the reduction of mesoporous pores along with increased macropores and wall thickness. As expected, SEM analysis revealed a growing number of 500 nm - 2 μm cellular macropores with the increasing FT cycles and increased skeleton thickness from 40 nm (AP-FT-0) to over 200 nm (AP-FT-3). Nitrogen adsorption-desorption isotherms for AP-1 and AP-3 demonstrated that more freeze-thaw cycles led to decreased BET specific surface area from 154 $m^2 g^{-1}$ to 70 $m^2 g^{-1}$ as well as reduced mesoporous volume from 0.45 $cm^3 g^{-1}$ to 0.28 $cm^3 g^{-1}$, while the increase of pore size from 11 nm to 16 nm (Supplementary Fig. 16). These phenomena suggest that the freeze-thaw induced fiber-to-wall skeleton evolution makes the packing of fibers tighter, and the integrity and synergy of ANF and PVA more obvious, thus, resulting in further mechanical enhancement of AP aerogels (Supplementary Fig. 17)[45].

## Fabrication of aerogels via configuration editing

So far, by TC strategy, we have succeeded in controlling the sol-gel process, modulating mechanical properties of the gels, and improving toughness of aerogels, which lays the foundation for configuration editing, configuration locking and fabrication of tough aerogel with complicated configuration through soft to hard modulation. To be more specific, configuration editing is based on the flexibility of soft organogel, that is, the transformable gel precursor, which could deform and recover easily. Configuration locking relies on the mechanical difference between organogel and hydrogel, where the editing-induced stress stored in hydrogel is too small to rebound the deformation locked by mechanical enhancement from organogel to hydrogel. Finally, fabrication of tough aerogel with complicated configuration could be realized by solvent exchange and supercritical drying in sequence. The example of configuration editing, configuration locking, and the resulting tough aerogel with complicated configuration was shown in Fig. 4a. The rectangular transformable organogel precursor was firstly twisted into a tortile strip with spring-

back property, empowering the repeatable configuration editing. Then after being immersed in water, the configuration of the distorted strip was locked to from tortile hydrogel strip in second coagulation. Finally, aerogel with both tortile configuration and high toughness was obtained by solvent exchange and supercritical drying in sequence.

In addition to configuration editing capabilities, the fidelity of configuration is another crucial parameter for evaluating the feasibility of TC strategy in ensuring consistency between the target and designed configurations. The precise tracking of configuration retention and distortion before/after each coagulation step was conducted. In freeze-thawing processes, no significant changes in configuration were observed, possibly due to the inhibitory effect of the gel network on continuous growth of DMSO crystals. During solvent exchange processes in the second coagulation step, a volume shrinkage of approximately 40% occurred (Supplementary Fig. 18) Even so, the overall configuration of the gel remained unchanged. For instance, after secondary coagulation, bending angles such as 90°, 135°, and 180° in a bent rectangular spline remained unaltered (Supplementary Fig. 19). Similarly, samples with twisting and twining editing retained the oringinal configurations after secondary coagulation (Supplementary Fig. 20). Therefore, benefiting from its high fidelity towards maintaining configurations during processing steps, TC strategy enables obtaining aerogels with desired target configurations.

Benefiting from the mechanical enhancement and configuration editing capabilities conferred by the TC strategy, tough aerogels with various complex configurations could be successfully fabricated. Typically, aerogels with complex structures can be prepared by clay sculpture, origami, weaving knot, etc., inspired by traditional folk arts (Fig. 4b, c). The bowl configuration hydrogel was prepared from three-dimensional gel by clay molding process. The two-dimensional gel film is folded into a boat through the origami process, and the one-dimensional gel string is woven into a Chinese knot through the weaving process. It's worth noting that despite repeated gel folding during the configuration editing process, no significant cracks were observed in the aerogel structure because of the rebound resilience (transformable capability) of organogel (Supplementary Fig. 21a). However, it is not feasible to edit the configuration of tough aerogels directly, which would crease or even fracture after bending or twisting (Supplementary Figs. 21b, 22). This fully highlight the advantages of TC strategy in configuration editing of tough aerogels.

Configuration editing capability endowed from TC strategy has provided great opportunities to break through performance limits and expand application areas of aerogels. Just with a few simple configuration-editing steps, such as bending, twisting, spiraling, folding, and knotting, aerogels with complex configurations like wave, helix, spring, and 3D hollow structure can be designed to meet specific needs and break through the mechanical constraints of aerogels (Fig. 4d–f, Supplementary Figs. 23–25). For example, differing from irreversible compressive deformation of aerogel monolith, the AP aerogel with wavy configuration accomplished 25% compressive resilience (Supplementary Figs. 25, 26). The aerogel springs could deform freely with good stretching, bending, and twisting resilience (Fig. 4d, Supplementary Fig. 27). More elaborately, the configuration parameters can be fine-tuned to effectively control the performance, such as diameter and height of the pitch, and winding method of the aerogel curly structure. By tuning the ratio of diameter ($D$) to height ($h$) of the pitch, various aerogel springs with different $D/h$ ratios were prepared and the tensile property was tested (Supplementary Fig. 28). With the increase of $D/h$ value from 1.0 to 3.0, -300% to 900% breaking elongation ratio were regulated (Fig. 4e). Though changing the helically winding method of aerogel spring with constant radius and variable height, a new curly configuration, named aerogel coil, could be obtained by rotating AP organogel splines concentrically with constant height and variable radius (Supplementary Fig. 29). The stretched coil aerogel could bounce back quickly (Fig. 4f), with at least 90% resilience

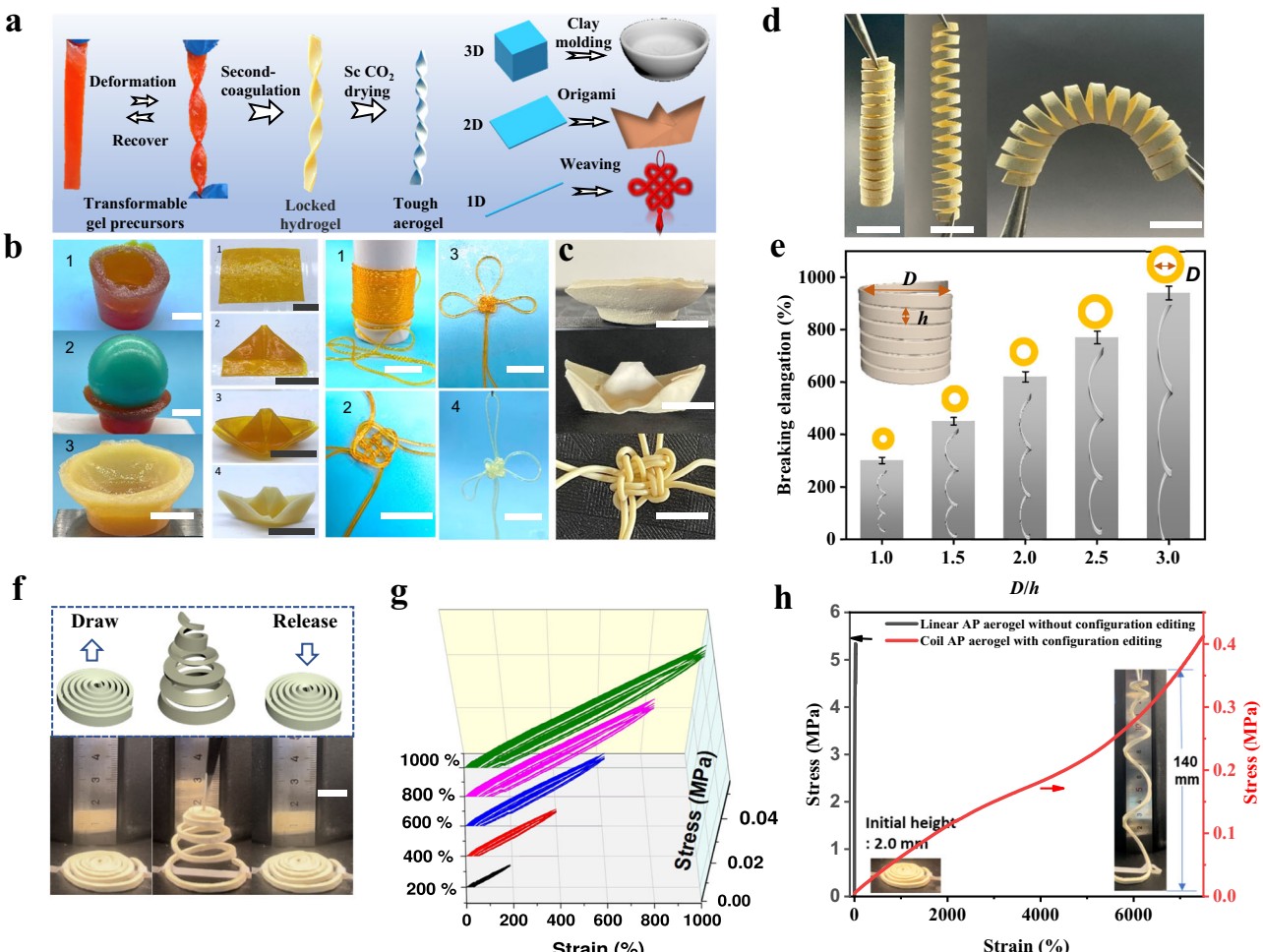

**Fig. 4 | Fabrication of aerogels via configuration editing. a** Schematic diagram of configuration editing, configuration locking, and aerogel preparation. **b** AP gel editing process inspired by folk arts, including clay sculpture, origami, and weaving. Scale bar: 3 cm. **c** Photos of prepared aerogels with complex configurations inspired by folk arts. **d** aerogel spring to stretch and bend. Scale bar: 1 cm. **e** Tensile property of various aerogel springs with different $D/h$ ratios, where $D$ means diameter of spring, and $h$ means the height of pitch as illustrated. Error bars indicate mean ± SD. **f** Tensile resilience of coil aerogel. Scale bar: 1 cm. **g** Tensile cyclic curves of coil aerogels with 200%, 400%, 600%, 800%, 1000% strain. **h** Stress-strain curves of linear aerogel without configuration editing and coil aerogel with configuration editing by TC strategy.

rates after releasing from 200%, 400%, 600%, 800%, and even 1000% tensile strain as the tensile cycle curves illustrated (Fig. 4g, Supplementary Fig. 30). If stretching further, the tensile elongation of coil aerogel exceeds 7000%, far superior to that of linear aerogels by TC strategy with only 25% breaking elongation ratio (Fig. 4h). The above experiments indicated that TC strategy has a strong configuration editing ability for complex aerogel design and can effectively adjust the configuration parameters of aerogel, which is of great value for breaking through the mechanical limitations of aerogel and developing configuration-dependent functional materials.

**Performance of the configuration-edited aerogel**
Configuration-dependent functional products such as self-supporting insulation structures, were fabricated subsequently by AP tough aerogels. AP aerogel scaffold was successfully prepared as self-supporting insulation aerogel and supported 1 kg load easily without obvious deformation benefiting from the high mechanical strength (Supplementary Fig. 31). Then, three comparison experiments were conducted to evaluate the thermal insulation property of aerogel scaffold as follows (Fig. 5a). Between the samples and 150 °C hot table are AP-17 aerogel plate (thermal conductivity: 0.038 W m$^{-1}$ K$^{-1}$, AP-17 aerogel scaffold, and nothing, the temperature of which were 47.0 °C, 35.6 °C, and 67.0 °C, respectively after 30 min (Fig. 5a).

By contrast, aerogel scaffold provided better insulation by separating the heat source from the load, highlighting the necessity and importance of aerogel configuration design in thermal insulation structures.

Configuration-editable capacity expanded the application of aerogel from the traditional areas to new fields such as thermal management and shape memory devices. Due to the stretchable and compressible characteristic of aerogel spring, controllable thermal management can be carried out. Spring aerogel-wrapped heating wires are used to simulate a thermal management system. By stretching and compressing, the gaps between aerogel spring dilate to accelerate thermal diffusion, and shrink to reduce heat dissipation (Fig. 5b). Instead of being well applied in traditional thermal storage field, aerogel-based host-guest materials here were designed as shape memory devices by filling phase-change materials in the pores of the above resilient aerogels[51]. The principle of shape memory utilizes both the phase-change-induced modulus change and configuration-endowed resilience. Paraffin wax was selected as the ideal guest phase-change material with phase transition at 44.5 °C and 61.7 °C. During the cooling process, the paraffin solidified below 30 °C along with the exothermal process and melted above 70 °C with the endothermic process (Fig. 5c). Tough aerogel springs were chosen as the host materials because of the tensile and compressive resilience

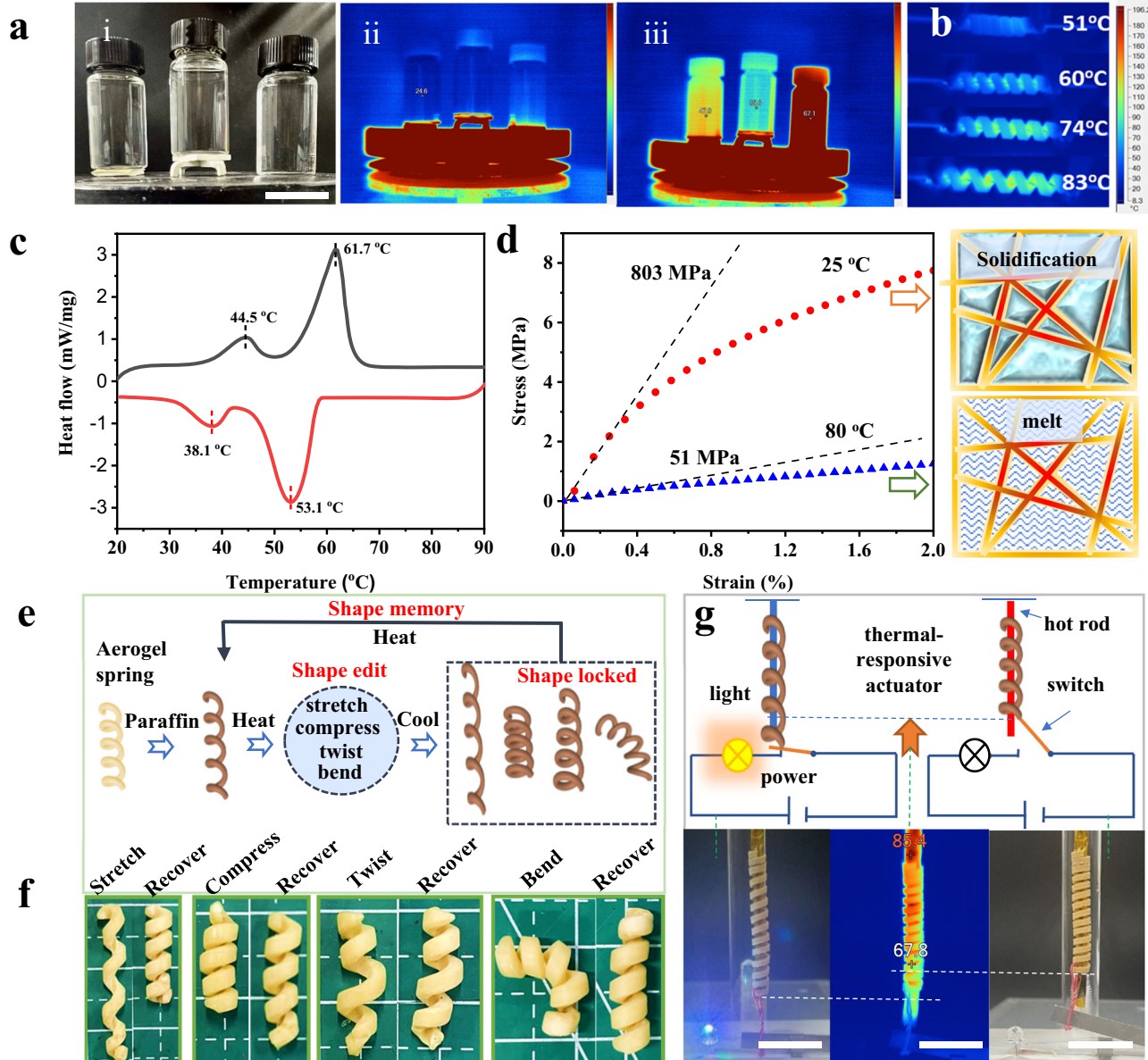

**Fig. 5 | Performance of the configuration-edited aerogel. a** Photographs and infrared images of different thermal insulation structures and thermal insulation effects (from left to right: AP−17 aerogel plate; AP−17 aerogel scaffold; nothing). Scale bar: 3 cm. **b** Aerogel spring used for thermal tuning. **c** DSC curve of the endothermic and exothermic process of paraffin wax. **d** Tensile strength of paraffin@ AP at 25 °C (solidification of paraffin) and 80 °C (melting of paraffin), respectively. Sample: rectangular Paraffin@AP spline with 2 mm × 2 mm cross section; initial loading gauge length: 10 mm, loading rate: 0.5 mm min⁻¹. **e** Schematic diagram of shape memory effect of paraffin@AP. **f** Paraffin@AP spring was deformed at high temperature and locked at low temperature, including stretching, compression, distortion, and bending; the configuration recovery at high temperature. **g** Paraffin@ AP aerogel spring used as the actuator for circuit switch. Scale bar: 3 cm.

mentioned above. Paraffin@AP was prepared by filling liquid paraffin wax in the aerogel at 80 °C. The successful fabrication of Paraffin@AP was confirmed by SEM, demonstrating the infiltration of paraffin into the pores of the cellular aerogel network (Supplementary Fig. 32). Tensile modulus of Paraffin@AP could be tuned by switching the solid/liquid state of the paraffin, which reached up to 803 MPa at 20 °C, and dropped to 51 MPa at 80 °C (Fig. 5d). The shape memory process was illustrated as follows. Paraffin@AP spring was firstly heated to 80 °C, and deformation of aerogel and inner network skeleton occurs under compression, stretching, twisting, bending, and other external forces. After cooling to 25 °C, the deformed aerogel and network skeleton are locked and the stress is stored inside the aerogel, due to the solidified paraffin and increased modulus. After heating to 80 °C, liquid paraffin loses its binding power and the stored stress will induce the elastic

recovery of aerogel and network skeleton, thus achieving shape memory effect (Fig. 5e, f, Supplementary Movie 1). At the same time, due to the hydrophobicity of paraffin, this device can also achieve shape memory in hot water. Paraffin@AP spring with 150% tensile strain bounced back to the original length within 10 s upon touching the hot water (Supplementary Fig. 33, Supplementary Movie 2). This host-guest shape-memory device has great potential in the field of actuators (Fig. 5g). For example, the circuit switch can be turned ON/ OFF by controlling the elastic recovery of stretching and bending recovery of Paraffin@AP spring (Fig. 5g, Supplementary Fig. 34). Predictably, by replacing the type of the guest phase change material and the configuration of the host tough aerogel, the stimulus mode and response condition can be regulated, thus expanding the family of shape memory device.

## Discussion

In conclusion, we have proposed and established an efficient TC strategy to fabricate configuration-editable tough AP aerogels. The mechanism investigation indicates that in the TC process, the flexible transformable oganogel precursors network formed by PVA after freeze-thawing in the first coagulation, and configuration-locking rigid hydrogel network generated due to the protonation of ANF and multiple intermolecular interactions between PVA and ANFs in the second coagulation. Through TC strategy, the gel network evolution from organogel, hydrogel to aerogel induced the mechanical enhancement significantly. Notably, compared with reported tough polymeric aerogels, AP aerogel has the highest mechanical properties, with a specific tensile modulus as high as 666 MPa cm$^3$ g$^{-1}$, and toughness as high as 2093 kJ m$^{-3}$, owing to the inner cellular 3D networks. More importantly, the TC strategy also endows the tough AP aerogels with configuration editing capability by oganogel editing, hydrogel locking and aerogel drying in sequence. For instance, configuration editing inspired by fork ats has guaranteed the successful fabrication of bowl, boat, and Chinese knot aerogel. By designing the configuration and tuning the parameters, aerogels with complex configurations, such as aerogel springs and coils, successfully broke through the mechanical constraints (e.g., tensile elongation varied from 25% to 7000%) and expanded application areas of aerogels in the fields of self-supporting thermal-insulation structures, thermal management devices, stimuli-responsive shape memory devices, etc. This study provides important insight for future design of configuration-editable aerogels and pushes forward the development of high-strength porous materials with special configurations. Although the TC strategy enable the aerogels with configuration editing capability, when applying it in commercial and engineering applications, there are still some challenges like the energy and time consumption issues in freezing-thawing process, sample volume shrinkage problems during solvent exchange and supercritical drying. We eagerly anticipate further breakthroughs in these areas in the future.

## Methods

### Materials

Kevlar 1000D was purchased from DuPont Company. PVA (Viscosity: 25.0–30.0 mPa.S; 97.5–99.0 mol% hydrolyzed) and Potassium hydroxide (KOH) was obtained from Shanghai Aladdin Biochemical Technology Co., Ltd. Potassium tert-butoxide (t-BuOK), Dimethyl sulfoxide (DMSO) was obtained from China National Pharmaceutical Group Co., Ltd. (Sinopharm). Ethyl alcohol was obtained from Suzhou Jingxie High and New Electronic Material Co., Ltd. Deionized water (18.2 MΩ cm$^{-1}$) was obtained from a Millipore-Q system. All other reagents were used without further purification.

### Preparation of ANF-PVA solution

Kevlar 1000D was dissolved in KOH/t-BuOK DMSO under magnetic stirring at 25 °C for 7 days to obtain a dark red, viscous solution of Aramid nanofibers (ANFs) (2.0 wt%). PVA was dissolved in DMSO (10 wt %) under magnetic stirring at 95 °C for 7 days. Mixing these two liquid precursors with 1:1 mass ratio leads to the precursor of 6.0 wt% ANF-PVA solution. Precursors with 5.5 wt%, 5.0 wt%, 4.5 wt% and 4.0 wt% were obtained by adding additional pure DMSO to the mixture, retaining the mass ratio between dispersed ANFs and dissolved PVA at 1:5. The mass percentage means the mass proportion of PVA and ANF together to the total mass of the solution.

### Preparation of AP organogel

The liquid mixture of ANF-PVA was poured into a mold or casted on a flat PET film and seal off water in the air. The first coagulation of ANF-PVA mixtures was achieved by cooling in the refrigerator at -25 °C for 12 hours and thawing at room temperature. The number of freezing-thawing sessions can be adjusted according to actual needs. After freezing-thawing, red AP organogel were obtained and allow complex configuration editing.

### Preparation of AP hydrogel and aerogel

The second coagulation of ANF-PVA mixtures was achieved by solvent exchange in deionized (DI) water, leading to white solid hydrogels. The configuration editing in the first coagulation were locked in this process. The hydrogel samples were immersed and exchanged solvent in ethanol for times followed by supercritical drying to generate targeted aerogels.

### Scanning electron microscopy (SEM)

The morphology of the obtained aerogels was characterized by scanning electron microscopy (SEM, S-4800, Hitachi, Japan) at an acceleration voltage of 5 kV.

### Fourier transform infrared spectroscopy

The structure of the samples was determined by the Fourier Transform Infrared Spectroscopy (FTIR 5700, FL, USA) over 64 scans recorded with a resolution wavelength of 4 cm$^{-1}$.

### Pore size distribution and average pore diameter

were analyzed by the Barrett-Joyner-Halenda (BJH) nitrogen adsorption and desorption method (ASAP 2020, Micromeritics, USA), and the specific surface area and pore volume were determined by the Brunauer-Emmett-Teller (BET) method, based on the amount of N$_2$ adsorbed at pressure $0.05 < P/P_0 < 0.3$.

### Infrared thermal images

were taken by an Infrared camera (Escalab 250 Xi, Thermo Scientific, USA), the working distance was about 30 cm.

### UV−vis absorption spectrum

was tested by Jasco V-660 ultraviolet-visible spectrophotometer. Mechanical properties of the samples were conducted by an Instron 3365 tensile testing machine with a gauge length of 10 mm at a loading rate of 0.5 mm·min$^{-1}$. More than 3 samples were test for each condition.

### X-ray diffraction (XRD) analysis

was carried out with a Bruker D8 Advanced spectrometer, using Cu Kα radiation at 40 kV and 40 mA. The scattering angle (2θ) was ranged from 5° to 60°.

### Thermal stability

was determined by the thermal gravimetric analyzer (TGA, 209F1, NETZSCH, Germany) with the heating rate of 10 K min$^{-1}$ in a nitrogen atmosphere.

### Raman spectra

were recorded on a LabRam HR Raman spectrometer with 50 W He-Ne laser operating at 632 nm with a CCD deter.

## Data availability

The data supporting the findings are provided within this Article and its Supplementary Information are available from the corresponding author on request. Source data are provided with this paper.

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

## Acknowledgements

We are grateful for the support from the National Natural Science Foundation of China (Project 52203021 to L.L.; 52173052 to X.Z.; 52003290 to J.L.; 22275207 to Z.S.) and the Natural Science Foundation of Jiangsu Province (Project BK20220296 to L.L.; BK20211099 to Z.S.).

## Author contributions

X.Z., F.M. and L.L. conceived the idea and designed the experiments. X.Z. and F.M. supervised the project. L.L.and G.Y. conducted the experiments. The data were analyzed and processed by X.Z., L.L., G.Y., J.L., Z.S. L.L. prepared the manuscript and X.Z., J.L., Z.S. contributed to the revision.

## Competing interests

The authors declare no competing interests.
