## [Peer review file · Nature Communications]

REVIEWER COMMENTS

Reviewer #1 (Remarks to the Author):

The manuscript titled "Folk Arts-Inspired, Twice-Coagulated, Configuration-Editable Tough Aerogels Enabled by Transformable Gel Precursors," written by Li and co-workers reports a novel strategy to fabricate aerogels of arbitrary configuration by employing so-called twice-coagulated (TC) strategy. Using aramid nanofibers (ANF) and polyvinyl alcohol (PVA) as main components, an organogel is first formed by freeze-thawing the dimethyl sulfoxide (DMSO) solution (first coagulation), whereas the organogel provides a soft and malleable gel. Origami, weaving, and other fabrication methods were applied in this state. When the shaping is done, the pre-formed precursor is immersed in deionized water for solvent exchange, resulting in the second coagulation step to fix the morphology. Finally, solvent exchange in ethanol followed by supercritical drying produced aerogels in arbitrary shape.

The strategy is simple, smart, and intuitively well-acceptable. And, to the best of my knowledge, the TC strategy has appreciable novelty. The authors captured the interdisciplinary perspective of the manuscript effectively by (i) comparing different gels at various stages and levels of quality using materials engineering techniques and (ii) showing a wide range of proof-of-concept demonstration of applications. In general, I am supportive of the eventual publication of the manuscript based on the novelty and interdisciplinary application possibility. However, some key details that enables the successful fabrication are either missing or unexplored in the study. Some scientific descriptions in the manuscript are rather obscure or incorrect. These details must be rectified before the manuscript can be considered for publication.

(1) The degree of warping, shape retention, and distortion before/after each coagulation steps must be revealed precisely. It is well known that the shape fidelity is the challenge in freeze-thawing and solvent exchange processes, and thus the related details must be reported in the publication. The authors mentioned that cracking may occur when a rectangular aerogel is bent to change its shape; this statement is insufficient for the readers to pursue reproduction.

(2) The TC strategy is smart and interesting, but it necessitates lengthy production steps and shape fidelity issues during the production. The authors may elaborate the challenges in applying the TC strategy in commercial and engineering applications.

(3) It seems that the authors could achieve some 'sweet-spot' in terms of fabricating malleable organogel after the first coagulation step (freeze-thaw of the DMSO solution). How general is this strategy successful? What happens when ANF-PVA ratio changes and/or when freeze-thaw steps have different thermal history? How long is the organogel malleable yet formative (i.e., the effect of solvent drying in ambient condition)? Any systematic studies in these regards?

(4) Figure 1a – the cartoons have unexplained color schemes and unphysical squiggly shapes.

(5) Supplementary Figure 5 on hydrogen bonding. The cartoons are unphysical and not informative. Can a statistically significant amount of polymer segments ever line up as depicted in the figures? The figure must be improved to reflect reality with referencing serious studies on hydrogen bonding of PVA, ANF, and ANF-PVA.

(6) Related to the point (5), the existence of hydrogen bonding, the text about the Raman spectra (why it was explored, what we are supposed to see, and what we are actually seeing in the spectra) must be improved for better scientific description.

(7) Supplementary Figure 9a vs 9b – the comparison is not fair. The same type of folding (i.e., the inner corner of the folded area) must be compared.

- (8) Supplementary Figure 16 – explain the equation.
- (9) Figure 3g. The hysteresis curve never goes back to zero for higher deformation. In this circumstance, how can the multiple stress-strain curves line up nicely as shown in the figure? The details of the measurement, including the definition of 'zero' for multiple steps, must be described precisely.
- (10) In Introduction, "shape memory materials (SMPs)" seems weird as an acronym. The sentence may be revised to make sense.
- (11) In introduction, "Biomedical science" does not seem to be a parallel field as other listed fields (i.e., way too broad and non-descriptive).
- (12) On page 4, what is meant by "unedited regular aerogel"? Was it created through general polymerization without undergoing the TC strategy? Does it differ in configuration from the coil shape?
- (13) In reference to page 6, the author mentions a slight fading of color and a decrease in the UV absorption spectrum in Figure 1c, indicating that ANF did not participate in the formation of the organogel network in the first coagulation step. Can you provide further explanation for readers who may find this part confusing?
- (14) There seems to be a discrepancy between the "Kevlar fibers" shown in Figure 1b and the protonation of ANF mentioned on page 6. Can you clarify the relationship between the two? Or change the terminology to keep it consistent.
- (15) On page 7, the theoretical mass ratio of PVA to ANF was calculated as 5/1 using a weight loss value. While this makes sense, it may be unreasonable if heating above 500 degrees is not taken into account. Could you consider other factors such as the gel formation process or the contribution ratio of each component, such as cross-linking, to determine a more appropriate ratio? Alternatively, could you provide some guidance on how to adjust the ratio based on the intended application of the gel?
- (16) Building on the point (12), the comparison between the new aerogels and regular aerogels on page 14 and Figure 3h lacks information about the regular aerogels. Could you provide more details on their shape and fabrication method, such as whether they are rectangular or conventionally coiled without TC strategy?
- (17) There are a few typos in the manuscript, such as "fellow" instead of "follow" on page 16.

Reviewer #2 (Remarks to the Author):

In this paper, the authors developed a twice-coagulated (TC) strategy to fabricate the configuration-editable tough aerogels composed of aramid nanofibers (ANFs) and polyvinyl alcohol (PVA). The authors further performed a series of characterization for microstructures and mechanical properties of ANF-PVA (AP) tough aerogels. The results showed that AP aerogels have high both specific modulus and toughness. These excellent mechanical properties of AP aerogels are attributed to the TC processing and the inner cellular 3D networks. The authors also demonstrated some applications of configuration-editable aerogels, such as self-supporting thermal insulation structures, thermal management devices, and shape-memory devices. Overall, the manuscript is well-written and comprehensive. The study results would have significant impact on the design and fabrication of novel aerogels with excellent mechanical properties and remarkable configuration-editable capacity. But the authors have to address the following issues before the current manuscript can be recommended for publication,

1. Figure 1f shows the SEM image of AP-23 aerogel sample, indicating the cellular structures. The mechanical properties of aerogel samples are determined by such cellular structure. The authors are suggested to add some statements to describe the pore size and associated distributions, and thickness and composition of cell wall. Furthermore, the authors, on page 11, stated the influences of freeze-thaw cycle on the cellular structure to explain the enhancement of mechanical properties. The authors should add more detailed descriptions how the freeze-thaw cycle affects the cellular structure, especially the influences of the freeze-thaw cycle on the characteristic size (such as wall thickness and pore size), which is crucial to determine the mechanical properties.
2. Figure 2c-d shows the mechanical properties of organogel, hydrogel and aerogel. It is seen in these two figures that both modulus and strength of sample increase significantly with the evolution, but the elongation is reduced. It to some extent implies the conflict between strength and toughness. The authors should mention the reduction in the elongation of materials and further explain the relevant origin of such reduction.
3. Figure 2e-f shows the compressive and tensile stress-strain curves of AP aerogels with different densities. It is seen in Fig. 2e-f that the strength of AP aerogel increases with the increasing of density. Can the authors further quantitatively investigate the dependences of compressive and tensile strengths on the density and provide the relevant explanations for such dependence?
4. Figure 2a shows the comparison in specific modulus and toughness between AP aerogel and other aerogels. The authors are suggested to further compare the strength between AP aerogel and other aerogels, because the strength is generally a more important mechanical property compared with specific modulus in most practical applications.
5. Figure 3g shows the tensile cyclic curves of coil aerogels under different strains. It is seen in Fig. 3g that for the larger applied strains, there exists a hysteresis in the stress-strain curves. The authors should add some statements to discuss such hysteresis.
6. The mechanical properties and performances of aerogels are associated with the hybrid networks composed of ANF and PVA. The authors are suggested to add more statements to describe the formation of hybrid networks of ANF and PVA, especially the cross linking among different polymer chains. Recent experimental study (*Materials Today*, 2023, doi: 10.1016/j.mattod.2023.07.020) showed that after introducing the low content of ANF (about 0.1-0.3 wt%), various mechanical properties (including modulus, strength, fracture toughness and fatigue resistance) of 3D printable hydrogel have been improved at least 10 times. In the current study, the content of ANF is up to 2 wt%. The authors are suggested to add some discussions about the mechanical enhancement from ANF with different contents by referring the reference mentioned above.
6. In the current manuscript, the authors defined "AP-x" in two different ways. The first one is on page 5, where x means the density; the second one is on page 10, where x means the number of freeze-thaw cycles. The authors should modify these abbreviation meaning to distinguish them.
7. On page 15, the authors mentioned the measurement of tensile modulus of Paraffin@AP. The authors should provide more information (such as sample shape, loading conditions, loading rate) about the tensile tests of Paraffin@AP.

Reviewer #3 (Remarks to the Author):

Configuration editing is critical for the practical and flexible application of aerogels. However, this is largely hindered by the high porosity and weak mechanical strength of aerogels. In this study, Li et al. tried to address this challenge by establishing an unique twice coagulation (TC) strategy through the use of transformable gel precursors. This endeavor is conceptually new, which points out a new direction for aerogel fabrication and should greatly advance the controlled fabrication of aerogels with tailored properties/applications. Also, the manuscript is well organized and the conclusion is well supported by the experiments/analysis. After minor revision as shown as follows, I recommend the publication of this paper in *Nature Communications*.

1. The detailed information of TGA and Raman scattering spectra should be provided in Characterization part.
2. Schematic diagram was recommended to intuitively represent the difference between cellular

network and 3D fiber network.

3. It is better to provide evidence to prove the successful fabrication of Paraffin@AP.

4. Some format errors should be carefully checked, such as spaces between numbers and units.

For Reviewer #1:

Comments:

The manuscript titled "Folk Arts-Inspired, Twice-Coagulated, Configuration-Editable Tough Aerogels Enabled by Transformable Gel Precursors," written by Li and co-workers reports a novel strategy to fabricate aerogels of arbitrary configuration by employing so-called twice-coagulated (TC) strategy. Using aramid nanofibers (ANF) and polyvinyl alcohol (PVA) as main components, an organogel is first formed by freeze-thawing the dimethyl sulfoxide (DMSO) solution (first coagulation), whereas the organogel provides a soft and malleable gel. Origami, weaving, and other fabrication methods were applied in this state. When the shaping is done, the pre-formed precursor is immersed in dionized water for solvent exchange, resulting in the second coagulation step to fix the morphology. Finally, solvent exchange in ethanol followed by supercritical drying produced aerogels in arbitrary configuration.

The strategy is simple, smart, and intuitively well-acceptable. And, to the best of my knowledge, the TC strategy has appreciable novelty. The authors captured the interdisciplinary perspective of the manuscript effectively by (i) comparing different gels at various stages and levels of quality using materials engineering techniques and (ii) showing a wide range of proof-of-concept demonstration of applications. In general, I am supportive of the eventual publication of the manuscript based on the novelty and interdisciplinary application possibility. However, some key details that enables the successful fabrication are either missing or unexplored in the study. Some scientific descriptions in the manuscript are rather obscure or incorrect. These details must be rectified before the manuscript can be considered for publication.

Response: We appreciate the reviewer's positive evaluation of our manuscript. We thank the reviewer very much for the constructive comments and suggestions to improve our manuscript greatly. We have addressed all the issues in a point-by-point manner and all the changes to the original manuscript are highlighted in blue.

Question 1. The degree of warping, configuration retention, and distortion before/after

each coagulation steps must be revealed precisely. It is well known that the configuration fidelity is the challenge in freeze-thawing and solvent exchange processes, and thus the related details must be reported in the publication.

The authors mentioned that cracking may occur when a rectangular aerogel is bent to change its configuration; this statement is insufficient for the readers to pursue reproduction.

Response: Thanks for your this professional constructive advice. Based on this suggestion, AP organogels with different configurations and sizes (e.g., monolith, film, and strip gels) were prepared and the degree of warping, configuration retention, and distortion during each coagulation process steps have been revealed precisely.

More statement about the detail of configuration editing has been provided and a diagram was added to make it easier to understand and reproduce.

Revision: Concise description about configuration fidelity was added in **last paragraph of “Mechanical properties of AP organogel, hydrogel and aerogel.”**

“In addition to configuration editing capabilities, the fidelity of configuration is another crucial parameter for evaluating the feasibility of TC strategy in ensuring consistency between the target and designed configurations. The precise tracking of configuration retention and distortion before/after each coagulation step was conducted. In freeze-thawing processes, no significant changes in configuration were observed, possibly due to the inhibitory effect of the gel network on continuous growth of DMSO crystals. During solvent exchange processes in the second coagulation step, a volume shrinkage of approximately 40% occurred (**Supplementary Fig. 16**) Even so, the overall configuration of the gel remained unchanged. For instance, after secondary coagulation, bending angles such as 90°, 135°, and 180° in a bent rectangular spline remained unaltered (**Supplementary Fig.17**). Similarly, samples with twisting and twining editing retained the original configurations after secondary coagulation (**Supplementary Fig. 18**). Therefore, benefiting from its high fidelity towards maintaining configurations during processing steps, TC strategy enables obtaining aerogels with desired target configurations.(**Line 23, Page 15**)

Supplementary Fig. 16. Volume shrinkage during twice coagulation process.

Supplementary Fig. 17. Shape retention of the bending gels. The bending angle of 90°, 135°, 180° of the bended rectangular spline remains unchanged after secondary coagulation. Scale bar, 1cm.

Supplementary Fig. 18. Shape retention of the twisting and twining gels. The organogels with twisting and twining editing could also keep the original configurations after secondary coagulation.

Detailed description of configuration editing could be seen below **Supplementary Fig. 20** in the Supplementary Information.

“The cracking would occur if directly editing aerogel such as stretching, compression, bending, and twisting aerogel. For example, in **Supplementary Fig. 20**, when winding the rectangular spline with 2 mm×2 mm cross section along a cylinder with 8 mm diameter and unwinding outward, the aerogel spline is cracked.”

Supplementary Fig. 20

Question 2. The TC strategy is smart and interesting, but it necessitates lengthy production steps and configuration fidelity issues during the production. The authors may elaborate the challenges in applying the TC strategy in commercial and engineering applications.

Response: Thank you for your nice suggestion. Indeed, lengthy production steps and configuration fidelity issues should be considered during the production of aerogel. Thus, we have discussed this challenge for future study.

Revision: These challenges have been added to the “**Discussion**” section.

“In conclusion, we have proposed and established an efficient TC strategy to fabricate configuration-editable tough AP aerogels... Although the TC strategy enable the aerogels with configuration editing capability, when applying it in commercial and engineering applications, there are still some challenges like the energy and time consumption issues in freezing-thawing process, sample volume shrinkage problems during solvent exchange and supercritical drying. We eagerly anticipate further breakthroughs in these areas in the future.” (Line 20, Page 23)

Question 3. It seems that the authors could achieve some ‘sweet-spot’ in terms of fabricating malleable organogel after the first coagulation step (freeze-thaw of the DMSO solution). How general is this strategy successful? What happens when ANF-PVA ratio changes and/or when freeze-thaw steps have different thermal history? How long is the organogel malleable yet formative (i.e., the effect of solvent drying in ambient condition)? Any systematic studies in these regards?

Response: Thank you for raising this important point. Actually, ANF-PVA ratio, thermal history, and air humidity will affect the malleability of organogel. We did systematic studies in these regards before, and the related detailed discussion have been added in the revised manuscript.

Revision: At the beginning of “**Twice-coagulated (TC) strategy**” part in the main text, these influencing factors are summarized.

“**Twice-coagulated (TC) strategy.** The specific twice-coagulated process and the network transition mechanism were primarily investigated. First of all, to determine the optimal experimental conditions, a comprehensive investigation and detailed discussion on the influence of ANF-PVA ratio, freezing to thawing cycles, and ambient conditions on the TC process were conducted and presented in the Supplementary Information (Supplementary Fig. 1-5). The optimal organogel-hydrogel-aerogel transition was performed as follows: firstly, 4.0 wt%, 4.5 wt%, 5.0 wt%, 5.5 wt%, 6.0 wt% ANF-PVA mixed solutions in alkali (KOH or t-BuOK) DMSO with 5:1 mass ratio of PVA to ANF were prepared. In the first coagulation setp, the mixed solutions were coagulated into red ANF-PVA composite organogels (AP organogels) after three freeze-thawing cycles, which need to be sealed to avoid humidity absorption from ambience.” (Line 6, Page 5)

We then discussed in detail the influence of different factors on the gel malleability, including 1) the ratio of ANF to PVA, 2) thermal history, and 3) ambient condition in in revised **Supplementary Note 1.**

1) The ratio of ANF to PVA. To investigate the impact of the ANF to PVA ratio on the malleability of organogels, we examined the mechanical strength of AP organogels at various ANF/PVA ratios. The experimental design was: initially, different ratios of ANF and PVA (ANF/PVA=1/80, 1/40, 1/20, 1/10, 1/5, 1/2.5, and 1/1.25) were mixed. Supplementary Fig. 1 shows that gel formation occurred immediately after mixing when the ANF/PVA ratio was below 1/20 or above 1/1.25. At a ratio of 1/2.5, heterogeneous solutions with noticeable small gel particles were observed. The presence of gel monoliths and small gel particles indicated inadequate mixing between ANF and PVA at these ratios, which were not suitable for TC strategy. After conducting more detailed experiments, we finally confirmed that elastic organogels could be formed within the range of ANF to PVA ratios from 1/15 to 5/1. When the ANF to PVA ratio exceeded 1:5 or fell below 1:15, insufficient mixing or excessive gelling occurred and the resulted organogels are too weak to be edited after freeze-thaw treatment (**Supplementary Fig. 2**). The possible reason is that excessive PVA content leads to protonation of ANF by hydrogen on the PVA, thereby promoting the formation of ANF gels. Conversely, an excessive amount of ANF induces gelation of PVA at room temperature due to the presence of excess alkali in ANF. Moreover, excessive ANF affects the crosslinking of PVA gels by wrapping or obstructing PVA molecular chains, consequently reducing the strength of organogels.

Subsequently, we conducted further investigations on the mechanical strength of organogels and hydrogels with an ANF/PVA ratio ranging from 1/15 to 1/5 (**Supplementary Fig. 3**). The strength of organogels increases gradually with the increase of ANF content. When the ANF/PVA ratio increases from 1/15, 1/10 to 1/5, the compression modulus increases from 5 kPa, 9 kPa to 14 kPa. The tensile elongation at break was significantly enhanced from 45% and 73% to 110%. This improvement can be attributed to the proportional increase in the number of cross-linking points of PVA induced by the ANF. Similarly, the strength of hydrogels exhibited a positive correlation with ANF content, owing to the formation of a highly cross-linked hybrid network between ANF and PVA that effectively transfers applied stress within the hydrogel. The higher the content of ANF, the higher the crosslinking density and

mechanical strength. Different from organic gels, the elongation at break of hydrogel decreases with the increase of ANF content. This is because in organogels with low crosslinking densities, the increasing crosslinking density can better resist stress and reduce the risk of fracture. While in hydrogel with high crosslinking densities, as the crosslink density increased, the elastic PVA molecular chain length between crosslinkers decreased. According to rubber-like elasticity theory, the extensibility of elastomers is generally proportional to the number of monomer units between crosslinkers (Matter **2022**, 5, 237–252; Chem. Rev. **2021**, 121, 4309). Therefore, the elongation at break of the AP hydrogel was reduced with the ANF content increase. (Mater. Today **2023**, 68, 84-95)

Supplementary Fig. 1. The mixture of ANF and PVA with different ratios (ANF/PVA=1/80, 1/40, 1/20, 1/10, 1/5, 1/2.5, 1/1.25 respectively)

Supplementary Fig. 2. The mixture of ANF and PVA and the configuration editability.

Supplementary Fig. 3 The compressive and tensile stress-strain curves of AP organogels (a, b) and hydrogels with different ratio of ANF/PVA (c, d).

2) **Thermal history.** To study the influence of the thermal history of ANF to PVA on malleable organogel, the mechanical strength of AP organogel by freezing-thawing with different times was tested respectively. The mechanical strength increases gradually with the increase of freezing-thawing times. (Supplementary Fig. 4.)

Supplementary Fig. 4. The tensile stress-strain curves of AP organogels with different freeze-thaw times (the ratio of ANF/PVA is 1/5).

3) Ambient condition. The organogel is still malleable within 2 hours in ambient condition, however, it will lose its formative nature for a long time. Solvent drying is an inevitable process for most gels; however, in ambient conditions, here the absorption of humidity emerges as another influential factor. The weight of the organogel continues to increase in ambient conditions (25°C, 50% relative humidity), as demonstrated in **Supplementary Fig. 5**. This observation indicates the absorption of humidity from the air, ultimately leading to ANF protonation and rendering the organogel non-malleable. Consequently, it is necessary to store organogels under sealed conditions.

Supplementary Fig. 5. The weight variation of AP organogels exposed in ambient condition (25 °C, relative 50% humidity). W_0 is the initial weight of sample before exposing in ambient condition, and W_t is the real-time weight of sample after exposing in ambient condition.

Question 4. Figure 1a – the cartoons have unexplained color schemes and unphysical squiggly configurations.

Response: Thanks for your raising this point. We have redesigned and improved Figure 1a, color schemes were explained carefully listed at the bottom of the figure and the unphysical squiggly configurations were corrected.

Revision: The improved figure could be seen in the revised **Figure 1a**.

Fig. 1a

Question 5. Supplementary Figure 5 on hydrogen bonding. The cartoons are unphysical and not informative. Can a statistically significant amount of polymer segments ever line up as depicted in the figures? The figure must be improved to reflect reality with referencing serious studies on hydrogen bonding of PVA, ANF, and ANF-PVA.

Response: Thanks for your suggestion. As shown in **Supplementary Scheme 1**, we redrew the chemical structure and the hydrogen bonding between PPTA-PPTA, PVA-PVA and PPTA-PVA with referencing some important literatures related to hydrogen bonding of PVA, ANF, and ANF-PVA. (Nat. Commun., **2022**, 13, 4242; Adv. Mater. **2018**, 30, 1703343; Adv. Sci. **2020**, 1902740)

Revision: The revised figures could be seen in **Supplementary Scheme 1**.

Supplementary Scheme 1.

Question 6. Related to the point (5), the existence of hydrogen bonding, the text about the Raman spectra (why it was explored, what we are supposed to see, and what we are actually seeing in the spectra) must be improved for better scientific description.

Response: We hope to characterize the deprotonation/protonation state of ANF in organogels and hydrogels by Raman spectroscopy and explored the gel network evolution during twice coagulation.

Revision: The modified text can be observed in **Paragraph 2 and Paragraph 3 within the section titled "Twice-coagulated (TC) strategy"**.

“The Raman spectrum of ANF-PVA mixture was similar to aramid nanofibers deprotonated by KOH reported elsewhere, indicating the successful deprotonation of ANF. The Raman spectrum of AP organogel remained unchanged compared with that of ANF-PVA mixture, suggesting that ANF was still in a deprotonated state and made no contribution to the first coagulation, ensuring the construction of first PVA elastic network.” (Line 14, Page 6)

“To investigate the internal network of hydrogel during secondary coagulation, a series of tests and characterizations were conducted. After solvent exchange in water, the transformation of AP organogel into hydrogel induced a significant shift in the Raman spectrum within the range of 800-1800 cm^{-1} . This observed Raman spectrum exhibited

striking similarities to that of Aramid fibers, providing evidence for the protonation of ANF during the second coagulation process.” (Line 9, Page 7)

Question 7. Supplementary Figure 9a vs 9b – the comparison is not fair. The same type of folding (i.e., the inner corner of the folded area) must be compared.

Response: Thank you for your suggestion. To be more convincing, the same type of folding was compared.

Configuration edit by TC strategy: Organogel film - Forward folding (the inner corner of the folded area is zero) - Reverse folding (the inner corner of the folded area is zero) - Unfolding (for SEM observation) - Dry.

Configuration edit Directly: Aerogel film - Forward folding (the inner corner of the folded area is zero) - Reverse folding (the inner corner of the folded area is zero) - Unfolding (for SEM observation)

Revision: The revised **Supplementary Fig. 19** presents SEM images of aerogels subjected to the same folding process.

Supplementary Fig. 19.

Question 8. Supplementary Figure 16 – explain the equation.

Response: Thank you for your suggestion. We have explained the equation in the revised manuscript.

Revision: Detailed explanation could be seen below **Supplementary Fig. 26** in the

revised Supplementary Information.

“Theoretical ultimate tensile ratio: $f = (1 + \sigma) \pi D/h$. where σ is break elongation rate of linear aerogel, D is the diameter and h is the height of the pitch. The elongation is contributed by two parts, one is the elongation of the spring being straightened $\pi D/h$, and the other is the elongation at break of the straight aerogel $\sigma \pi D/h$.”

Question 9. Figure 3g. The hysteresis curve never goes back to zero for higher deformation. In this circumstance, how can the multiple stress-strain curves line up nicely as shown in the figure? The details of the measurement, including the definition of ‘zero’ for multiple steps, must be described precisely.

Response: Under large tensile strain, the deformation of aerogel includes elastic deformation of spring and intrinsic plastic deformation of ANF aerogel. Elastic deformation predominates as the primary mode of deformation, while plastic deformation acts as secondary one. Consequently, the tensile cycle curve exhibits near-linearity due to the dominant elastic spring-like behavior, whereas the hysteresis curve fails to return to zero owing to the presence of AP aerogel's intrinsic plasticity. Similar phenomena have been observed in many works of literature (Adv. Sci., **2023**, 10, 2300; Adv. Mater. **2018**, 30, 1703343). The measurement details were added to enhance understanding.

Revision: The complete stress-strain curves for higher deformation, along with the curve analysis and test conditions, are included in the **Supplementary Fig. 28**.

Supplementary Fig. 28. Complete tensile cyclic curves of coil aerogels with 1000% strain.

The details of the measurement: we define the “zero” as 0% strain. The stretching procedure is as follows: Each cycle starts with an initial 0% tensile strain, then stretches 20 mm /min to a preset tensile strain value (e.g. 1000%), then 20 mm /min returns to 0% tensile strain, and then repeat the process for the next cycle.

Question 10. In Introduction, “Shape memory materials (SMPs)” seems weird as an acronym. The sentence may be revised to make sense.

Response and Revision: Thank you for your reminding, we have revised it as “Shape memory materials (SMMs)”, referring the literature. (Mater. Today **2010**, 13, 54-61). SMPs is the abbreviation of “Shape memory polymers”. (Adv. Mater. **2021**, 33, 2000713) (Line 9, Page 2)

Question 11. In introduction, “Biomedical science” does not seem to be a parallel field as other listed fields (i.e., way too broad and non-descriptive).

Response and Revision: Thanks for your suggestion. We have deleted “Biomedical science”. (Line 19, Page 2)

Question 12. On page 4, what is meant by "unedited regular aerogel"? Was it created

through general polymerization without undergoing the TC strategy? Does it differ in configuration from the coil configuration?

Response: Unedited regular aerogel is also prepared by TC strategy, but without the configuration editing. For example, the difference of fabrication procedure between unedited regular aerogel and coil aerogel is as follow.

Unedited regular aerogel: first, the rectangular linear organogel splines were prepared by freeze-thawing of PVA and ANF mixture, then after solvent exchange in water, the linear organogel splines turned into hydrogel splines. Finally, the linear aerogel splines were obtained by subsequent solvent exchange in ethanol and supercritical drying.

Coil aerogel: first, the rectangular linear organogel splines were prepared by freeze-thawing of PVA and ANF mixture, then organogels splines wrapped around a cylinder to prepare coil organogels. After solvent exchange in water, the coil organogel splines turned into coil hydrogel splines. Finally, the coil aerogel splines were obtained by subsequent solvent exchange in ethanol and supercritical drying.

Revision: To be clear, “unedited regular aerogel” was modified with “linear aerogels by TC strategy” (Line 12, Page 4; Line 19, Page 18)

Question 13. In reference to page 6, the author mentions a slight fading of color and a decrease in the UV absorption spectrum in Figure 1c, indicating that ANF did not participate in the formation of the organogel network in the first coagulation step. Can you provide further explanation for readers who may find this part confusing?

Response: We have demonstrated the state of ANF during twice coagulation process by Raman spectroscopy. Here, color changes and UV absorption spectrum were used as more intuitive evidences for protonation/deprotonation state of ANF. To eliminate confusion, we provide further explanation in the revised manuscript.

Revision: The detailed explanation was provided in **paragraph 2** of the section titled "Twice-coagulated (TC) strategy".

“To enhance the intuitiveness of our findings, color changes and UV-Vis absorption spectra were employed as supporting evidence for the protonation/deprotonation state of ANF. Typically, in its deprotonated state, the ANF solution exhibits a red color and

displays a prominent absorption peak at 335 nm in the UV-Vis spectrum. Upon reprotonation, the solution's color transitions to a pale yellow while experiencing a significant decrease in absorption intensity. Comparative analysis of PVA-ANF mixed solutions revealed that even after undergoing freezing and thawing processes, the organic gel retained its red coloration and exhibited a similar UV absorption spectrum featuring a strong peak at 335 nm (Fig. 1c). These observations further support that ANF remained in a deprotonated state during the first coagulation step without participating in organogel network formation, consistent with Raman spectroscopy results. By the way, slight fading of color and reduction in UV absorption may be attributed to partial contact between ANFs repelled by DMSO crystals. So, the AP organogels could be structured as PVA organogel network containing un-fixed ANFs after the first coagulation step.” (Line 19, Page 6)

Question 14. There seems to be a discrepancy between the "Kevlar fibers" shown in Figure 1b and the protonation of ANF mentioned on page 6. Can you clarify the relationship between the two? Or change the terminology to keep it consistent.

Response: Thank you for your suggestion. Aramid nanofibers (ANF) is obtained and peeled from the Kevlar fibers (one kind of Aramid fibers) in KOH/t-BuOK DMSO solution as described in “Methods” section. To keep it consistent, we change the Kevlar fibers as Aramid fibers in **Figure 1b** and elsewhere in the text. (Line 13, Page 7)

Question 15. On page 7, the theoretical mass ratio of PVA to ANF was calculated as 5/1 using a weight loss value. While this makes sense, it may be unreasonable if heating above 500 degrees is not taken into account. Could you consider other factors such as the gel formation process or the contribution ratio of each component, such as cross-linking, to determine a more appropriate ratio? Alternatively, could you provide some guidance on how to adjust the ratio based on the intended application of the gel?

Response: Thanks for your professional suggestion. In fact, the weight loss above 500 degrees in our calculations was taken into account. In order to avoid misunderstanding, we have further explained the calculation formula $(75\%/90\%)/(8.4\%/50\%) \approx 5/1$.

When applying TC strategy to configuration editing, many factors should be taken into consideration to determine appropriate ratio of the PVA to ANF. For example, the elasticity and flexibility of organogels should be considered to ensure the feasibility of configuration editing. The configuration locking ability of the hydrogel needs to be considered to ensure that the target configuration is obtained. The strength of the gel is also an important consideration to ensure the toughness of the aerogel. Based on such guidance and a series of experiments with different ANF/PVA ratio as displayed in Comment 1's answer, ANF/PVA=1/5 was chosen as the most appropriate ratio in our article. We have added a sufficient discussion on the ratio of ANF/PVA in the revised manuscript.

Revision : Calculation formulas and detailed explanations are added to “**Supplementary Note 2. Calculation of PVA/ANF**” in Supplementary Information.

Calculation of PVA/ANF. As TGA shown in figure 1e, the mass decomposed at 250-500 °C of pure PVA $m(PVA, 250-500\text{ °C})$ accounts for 90%, so the total mass of PVA is $m(PVA, 250-500\text{ °C})/90\%$. The mass decomposed at 520-620 °C of pure ANF $m(ANF, 520-620\text{ °C})$ accounts for 50%, so the total mass of ANF is $m(ANF, 520-620\text{ °C})/50\%$. The TGA curve of AP at 250 °C to 500 °C is attributed to the decomposition of PVA, and the decomposition mass is 75% $m(AP)$, so the total mass of PVA in AP is $75\% m(AP)/90\%$. Similarly, the decomposition curve of AP at 520-620 °C is attributed to the decomposition of ANF (PVA no longer decomposes at 520-620 °C), and the decomposition mass is 8.4% $m(AP)$, so the total mass of ANF in AP is calculated as $8.4\% m(AP)/50\%$. So the ratio of PVA/ANF is $[75\% m(AP)/90\%]/[8.4\% m(AP)/50\%]$, namely $(75\% / 90\%)/(8.4\% / 50\%)$. It can be expressed by the formula:

$$\frac{m(PVA)}{m(ANF)} = \frac{\frac{m(AP,250-500)}{90\%}}{\frac{m(AP,520-620)}{50\%}} = \frac{75\% m(AP)/90\%}{8.4\% m(AP)/50\%} = 5/1$$

The discussion on the ratio of ANF/PVA could be seen in Supplementary Note 1. in Supplementary Information

Question 16. Building on the point (12), the comparison between the new aerogels and

regular aerogels on page 14 and Figure 3h lacks information about the regular aerogels. Could you provide more details on their configuration and fabrication method, such as whether they are rectangular or conventionally coiled without TC strategy?

Response: Regular aerogel is prepared by TC strategy, but without the step of configuration editing. Their configuration was rectangular and the fabrication method was as follow: first, the rectangular linear organogel splines were prepared by freeze-thawing of PVA and ANF mixture, then after solvent exchange in water, the linear organogel splines turned into hydrogel splines. Finally, the linear aerogel splines were obtained by subsequent solvent exchange in ethanol and supercritical drying.

Revision: To be clear, “regular aerogel” on page 14 and Figure 3h was modified with “linear aerogels by TC strategy”

Question 17. There are a few typos in the manuscript, such as "fellow" instead of "follow" on page 16.

Response: Thank you for your kind reminding. We have carefully checked the manuscript and revised the typos, such as “fellow” – “follow”. (Line 1, Page 21)

For Reviewer #2:

Comment:

In this paper, the authors developed a twice-coagulated (TC) strategy to fabricate the configuration-editable tough aerogels composed of aramid nanofibers (ANFs) and polyvinyl alcohol (PVA). The authors further performed a series of characterization for

microstructures and mechanical properties of ANF-PVA (AP) tough aerogels. The results showed that AP aerogels have high both specific modulus and toughness. These excellent mechanical properties of AP aerogels are attributed to the TC processing and the inner cellular 3D networks. The authors also demonstrated some applications of configuration-editable aerogels, such as self-supporting thermal insulation structures, thermal management devices, and configuration-memory devices. Overall, the manuscript is well-written and comprehensive. The study results would have significant impact on the design and fabrication of novel aerogels with excellent mechanical properties and remarkable configuration-editable capacity. But the authors have to address the following issues before the current manuscript can be recommended for publication,

Response: We appreciate the reviewer's positive evaluation of our manuscript. We thank the reviewer very much for the constructive comments and suggestions to improve our manuscript greatly. We have addressed all the issues in a point-by-point manner and all the changes to the original manuscript are highlighted in blue.

Question 1. Figure 1f shows the SEM image of AP-23 aerogel sample, indicating the cellular structures. The mechanical properties of aerogel samples are determined by such cellular structure. The authors are suggested to add some statements to describe the pore size and associated distributions, and thickness and composition of cell wall. Furthermore, the authors, on page 11, stated the influences of freeze-thaw cycle on the cellular structure to explain the enhancement of mechanical properties. The authors should add more detailed descriptions how the freeze-thaw cycle affects the cellular structure, especially the influences of the freeze-thaw cycle on the characteristic size (such as wall thickness and pore size), which is crucial to determine the mechanical properties.

Response: Thank you for raising this important suggestion. We have added some statements to describe the pore size and associated distributions, and thickness and composition of cell wall in Figure 1f. We also give more explanation on how the freeze-thaw cycle affects the cellular structure, especially the influences of the freeze-thaw

cycle on the characteristic size (such as wall thickness and pore size).

Revision: The description about the pore size and associated distributions, and thickness and composition of cell wall was added in the revised **paragraph 2 of the section titled "Twice-coagulated (TC) strategy"**

“From cross-section SEM image, all the aerogels AP-17, AP-19, AP-23, AP-25 and AP-27 undergoing three freeze-thaw cycles show continuous porous cellular network skeleton composed of ANFs wrapped in PVA, with a wall thickness of about 40-250 nm and a macropore size of about 500 nm to 2 μm . (Line 6, Page 9)

The nitrogen adsorption/desorption isotherms of the AP-17, AP-23 and AP-27 aerogels show hysteresis loops at high relative pressure, indicating the existence of plentiful mesopores (Fig. 1g). For example, the average mesoporous pore size of AP-23 aerogel is calculated to be 16 nm, with the pore size distribution ranging from 2 nm to 26 nm. (Line 14, Page 9)

Detailed explanation about the influence of the freeze-thaw cycle on cellular structures were provided in the revised manuscript. (Paragraph 4, Section: Mechanical properties of AP organogel, hydrogel and aerogel.)

“The freeze-thaw induced fiber-to-wall skeleton evolution may be attributed to repeated DMSO crystal repulsion and cross-linking of PVA during multiple freeze-thaw process. Specifically, in the absence of freeze-thaw treatment, the initially evenly dispersed ANF nanofibers and PVA in the solution was transformed into AP-FT-0 aerogel with a uniform 3D fiber network. After freezing treatment, the ANFs and PVAs were squeezed around the DMSO crystals to form a wall, meanwhile part of the PVAs were cross-linked, thereby preserving the configuration of the wall. The remaining uncrosslinked PVA/ANF walls were re-dispersed into the solution during the thawing process. Consequently, we observed the coexistence of the 3D fiber network and the wall in AP-FT-0 aerogel. As the number of freeze-thaws increases, the DMSO crystals continue to push the ANF-PVA around, leading to generation of new walls. This resulted in fewer fibrillar 3D skeletons but an increasing presence of cellular 3D skeletons as evidenced by AP-FT-3 samples. In this process, the growth and repulsion of DMSO crystals

caused aggregation of nanofibers and PVA, resulting in the reduction of mesoporous pores along with increased macropores and wall thickness. SEM analysis revealed a growing number of 500 nm-2 μ m cellular macropores with the increasing FT cycles and increased skeleton thickness from 40 nm (AP-FT-0) to over 200 nm (AP-FT-3). Nitrogen adsorption-desorption isotherms for AP-1 and AP-3 demonstrated that more freeze-thaw cycles led to decreased BET specific surface area from 154 m²/g to 70 m²/g as well as reduced mesoporous volume from 0.45 cm³/g to 0.28 cm³/g, while the increase of pore size from 11 nm to 16 nm (**Supplementary Fig. 15**).” (**Line 6, Page 14**)

Question 2. Figure 2c-d shows the mechanical properties of organogel, hydrogel and aerogel. It is seen in these two figures that both modulus and strength of sample increase significantly with the evolution, but the elongation is reduced. It to some extent implies the conflict between strength and toughness. The authors should mention the reduction in the elongation of materials and further explain the relevant origin of such reduction.

Response: Thank you for your suggestion. We have mentioned the reduction in the elongation of materials and further explained the relevant origin of such reduction in the revised manuscript.

Revision: Relevant expressions are added in the revised text (**Paragraph 1, Section: Mechanical properties of AP organogel, hydrogel and aerogel.**)

“It is worth noting that the tensile strength of AP-23 organogel, hydrogel, and aerogel follows a similar increasing trend, while the elongation at break decreases. As depicted in Fig. 2d, the tensile moduli of AP-23 organogel, hydrogel, and aerogel increase from 0.3 MPa and 2.1 MPa to 87 MPa respectively; likewise, the breaking strength increases from 0.45 MPa and 1.1 MPa to 5.3 MPa accordingly; however, the elongation at break decreases from 110% and 60% to only 25%. This phenomenon can be attributed to variations in the crosslinking density. As mentioned above, organogels consist of flexible PVA networks with elastic PVA molecular chains and a low crosslinking density, enabling them to stretch extensively. The formation of a hybrid network between ANFs and PVAs in hydrogels leads to increased crosslinking densities, which subsequently reduces the elastic PVA molecular chain length between crosslinkers, thus reducing the

elongation at break of the AP hydrogel according to rubber-like elasticity theory (Chem. Rev. **2021**, 121, 4309; Mater. Today **2023**, 68, 84-95) Similarly, after drying in supercritical CO₂, the swelling and amorphous PVA molecular chains in wet gels undergo further crystallization and aggregation, resulting in an additional increase in cross-linking density and restricted movement of PVA molecular chains within aerogels. Consequently, aerogels exhibit the highest mechanical strength but the lowest elongation at break among these three gel states.” (Line 9, Page 11)

Question 3. Figure 2e-f shows the compressive and tensile stress-strain curves of AP aerogels with different densities. It is seen in Fig. 2e-f that the strength of AP aerogel increases with the increasing of density. Can the authors further quantitatively investigate the dependences of compressive and tensile strengths on the density and provide the relevant explanations for such dependence?

Response: We appreciate this important question. We have further quantitatively investigated the dependences of compressive and tensile strengths and provided relevant explanations for such dependence.

Revision: The quantitative investigation about the dependences of compressive and tensile strengths on the density was described in the main text (**Paragraph 2, Section: Mechanical properties of AP organogel, hydrogel and aerogel**)

“Like most aerogels, log–log plot of compressive and tensile modulus vs density is linear. (Chem. Mater. **2014**, 26, 4163–4171) For example, the tensile moduli of APs as a function of densities follow a relation $E \sim \rho^{3\pm 0.27}$, (E, Young’s modulus; ρ , density) (Supplementary Fig. 11), which is typical relation for cellular foam materials with a random network. (MRS Bull. **2003**, 28, 270–274)” (Line 14, Page 12)

The figures and relevant explanations were provided in the **Supplementary Figure 14**.

Supplementary Fig. 11. a) log–log plot of compressive modulus vs density (log standard deviation = 0.25, $R^2 = 0.99$). b) log–log plot of tensile modulus vs density (log standard deviation = 0.27, $R^2 = 0.97$).

This relationship between strength and density may be due to complex pore structure of AP aerogels, synergistic relationship between ANF and PVA, and self-organization behavior under stress. According to the theoretical analysis in the literature, the quantitative relationship between strength and density of cellular solid is different when the skeletons were in different configurations (MRS Bull. **2003**, 28, 270–274). AP aerogels have hierarchical pore structure, so the relationship between strength and density is nonlinear. In addition, the higher the density, the more obvious the synergistic effect of ANF and PVA and self-organization abilities, so the faster the increase of its mechanical strength. (Adv. Mater. **2018**, 30, 1703343; Nat. Commun. **2022**, 13, 4242)

Question 4. Figure 2a shows the comparison in specific modulus and toughness between AP aerogel and other aerogels. The authors are suggested to further compare the strength between AP aerogel and other aerogels, because the strength is generally a more important mechanical property compared with specific modulus in most practical applications.

Response: Thank you for your suggestion. We have further compared the strength between AP aerogel and other aerogels. The results were added in supplemental figure 12 in the Supplementary Information. Compared with other aerogels, AP aerogel still has remarkable high strength with the modulus of 180 MPa for AP-27

Revision: The comparison was added in the main text (**Paragraph 3, Section:**

Mechanical properties of AP organogel, hydrogel and aerogel) and Supplementary Fig.12.

“When compared to the high-strength polymeric aerogels reported so far (Supplementary Table 1), AP aerogel has remarkable strength with the modulus of AP-27 as high as 180 MPa (Supplementary Fig.12).” (Line 4, Page 13)

Supplementary Fig. 12. The comparison of tensile modulus between high-strength polymeric polymers.

Question 5. Figure 3g shows the tensile cyclic curves of coil aerogels under different strains. It is seen in Fig. 3g that for the larger applied strains, there exists a hysteresis in the stress-strain curves. The authors should add some statements to discuss such hysteresis.

Response: Under large tensile strain, the deformation of aerogel includes elastic deformation of spring and intrinsic plastic deformation of ANF aerogel. So, there exists a hysteresis in the stress-strain curves due to the plastic deformation of aerogel. Similar phenomena have been observed in many literatures. (Adv. Sci., **2023**, 10, 2300; Adv. Mater. **2018**, 30, 1703343). The details of the measurement were added for better understanding.

Revision: We have added these statements in **Supplementary Fig. 28**.

Supplementary Fig. 28 Complete tensile cyclic curves of coil aerogels with 1000% strain.

The details of the measurement: we define the “zero” as 0% strain. The stretching procedure is as follows: Each cycle starts with an initial 0% tensile strain, then stretches 20 mm /min to a preset tensile strain value (e.g. 1000%), then 20 mm /min returns to 0% tensile strain, and then repeat the process for the next cycle.

Question 6. The mechanical properties and performances of aerogels are associated with the hybrid networks composed of ANF and PVA. The authors are suggested to add more statements to describe the formation of hybrid networks of ANF and PVA, especially the cross linking among different polymer chains. Recent experimental study (Mater. Today, 2023, doi: 10.1016/j.mattod.2023.07.020) showed that after introducing the low content of ANF (about 0.1-0.3 wt%), various mechanical properties (including modulus, strength, fracture toughness and fatigue resistance) of 3D printable hydrogel have been improved at least 10 times. In the current study, the content of ANF is up to 2 wt%. The authors are suggested to add some discussions about the mechanical enhancement from ANF with different contents by referring the reference mentioned above.

Response: Thank you for your professional advice. As discussed in “Twice-coagulated (TC) strategy” part, after freeze-thawing in the first coagulation, the flexible PVA network formed and transformable organogel precursors were prepared. After solvent exchange in water in the second coagulation, hybrid networks containing multiple intermolecular interactions between PVA-PVA, ANF-ANF, PVA-ANF induced the

generation of configuration-locking rigid hydrogel network. To express this process more clearly, the formation of hybrid networks of ANF and PVA were described in detail.

We also add some discussions about the mechanical enhancement from ANF with different contents by referring the reference (Mater. Today 2023, doi: 10.1016/j.mattod.2023.07.020).

Revision: The formation of hybrid networks of ANF and PVA and the cross linking among different polymer chains were described in the main text (**Paragraph 3, Section: Twice-coagulated (TC) strategy.**)

“To investigate the internal network of hydrogel during secondary coagulation, a series of tests and characterizations were conducted. After solvent exchange in water, the transformation of AP organogel into hydrogel induced a significant shift in the Raman spectrum within the range of 800-1800 cm^{-1} . This observed Raman spectrum exhibited striking similarities to that of Kevlar fibers, providing evidence for the protonation of ANF during the second coagulation process. (Fig. 1b). Simultaneously, the color of the AP organogel changed from red to white and the UV absorption peak showed a substantial decrease (Fig. 1c), which further indicates the re-protonation process of ANF in this process. Reprotonation of ANFs leads to the elimination of repulsion between ANFs, resulting in subsequent ANF-ANF interactions facilitated by π - π interactions between aromatic rings and hydrogen bonding among amide groups. Additionally, similar to other reported studies on the regulation of intermolecular interactions or crystallization of PVA chains induced by solvent exchange⁴³, the solvent exchange in water also facilitates further enhancement of PVA-PVA chain interactions within the hydrogel. Compared with the weak diffraction near 20° in organogel, attributing to the crystallization in Kevlar nanofibers or PVA macromolecules, the stronger XRD diffraction and new forming peak near 40° indicate the further crystallization of PVA chains in hydrogel (Supplementary Fig. 3). What’s more, the solvent exchange in water not only induced the reprotonation of ANFs, the crystallization of PVA, but also influence the formation of ANF-PVA hydrogen bonding. In ANF-PVA mixture and AP organogel, DMSO, as a strong hydrogen bond

acceptor, inhibits hydrogen bond formation between PVA and ANF. Upon solvent exchange with water, a weaker hydrogen bond acceptor, the hydrogen-bond interaction generate between the hydroxyl groups on PVA and carbonyl groups on ANFs due to the energetically favorable OH \cdots O=C interaction (Angew. Chem., Int. Ed. **2002**, 41, 49) and the cooperativity of supramolecular interactions. Infrared absorption spectra provide insights into the ANF–PVA interactions. It could be seen that the C=O stretching vibration peak at 1651 cm⁻¹ moves to 1644 cm⁻¹ in infrared spectrum (Supplementary Fig. 4), indicating the existence of hydrogen bonds between ANF-PVA. Therefore, to summarize, the results revealed that in the process of secondary coagulation, multiple interactions among PVA-PVA, ANF-ANF, and ANF-PVA coexist and synergistically promote the formation of a dense cross-linked network as illustrated in Supplementary Scheme 1. " **(Line 9, Page 7)**

Discussions about the mechanical enhancement from ANF with different contents were added in **Supplementary Note 1.** and **Supplementary Fig. 3** in Supplementary Information.

Subsequently, we conducted further investigations on the mechanical strength of organogels and hydrogels with an ANF/PVA ratio ranging from 1/15 to 1/5 (**Supplementary Fig. 3**). The strength of organogels increases gradually with the increase of ANF content. When the ANF/PVA ratio increases from 1/15, 1/10 to 1/5, the compression modulus increases from 5 kPa, 9 kPa to 14 kPa. The tensile elongation at break was significantly enhanced from 45% and 73% to 110%. This improvement can be attributed to the proportional increase in the number of cross-linking points of PVA induced by the ANF content. Similarly, the strength of hydrogels exhibited a positive correlation with ANF content, owing to the formation of a highly cross-linked hybrid network between ANF and PVA that effectively transfers applied stress within the hydrogel. The higher the content of ANF, the higher the crosslinking density and mechanical strength. Different from organic gels, the elongation at break of hydrogel decreases with the increase of ANF content. This is because in organogels with low crosslinking densities, the increasing crosslinking density can better resist stress and

reduce the risk of fracture. While in hydrogel with high crosslinking densities, as the crosslink density increased, the elastic PVA molecular chain length between crosslinkers decreased. According to rubber-like elasticity theory, the extensibility of elastomers is generally proportional to the number of monomer units between crosslinkers (Matter **2022**, 5, 237–252; Chem. Rev. **2021**, 121, 4309). Therefore, the elongation at break of the AP hydrogel was reduced with the ANF content increase. (Mater. Today **2023**, 68, 84-95, i.e. ref. 47 in the manuscript)

Supplementary Fig. 3 The compressive and tensile stress-strain curves of AP organogels (a, b) and hydrogels with different ratio of ANF/PVA (c, d).

Question 7. In the current manuscript, the authors defined “AP-x” in two different ways. The first one is on page 5, where x means the density; the second one is on page 10, where x means the number of freeze-thaw cycles. The authors should modify these abbreviation meaning to distinguish them.

Response and revision: Thank you for your very careful review. The “AP-x” was uniformly defined as aerogel with certain density, where x means the density. Aerogels undergoing different freeze-thaw cycles were shortened as “AP-FT-x”. (Line 25, Page 13; Line 1, 2, 4, 9, 18, 22, 23, Page14)

Question 8. On page 15, the authors mentioned the measurement of tensile modulus of Paraffin@AP. The authors should provide more information (such as sample configuration, loading conditions, loading rate) about the tensile tests of Paraffin@AP.

Response: Thank you for your reminding. Measurement of tensile modulus of Paraffin@AP: Rectangular Paraffin@AP spline with 2 mm × 2 mm cross section was conducted by an Instron 3365 tensile testing machine with a gauge length of 10 mm at a loading rate of 0.5 mm·min⁻¹.

Revision: The tensile test information has been added in the caption of Figure 4d.

Sample: rectangular Paraffin@AP spline with 2 mm × 2 mm cross section; initial loading gauge length: 10 mm, loading rate: 0.5 mm min⁻¹. (**Line 7, Page 22**)

For Reviewer #3:

Comment:

Configuration editing is critical for the practical and flexible application of aerogels. However, this is largely hindered by the high porosity and weak mechanical strength of aerogels. In this study, Li et al. tried to address this challenge by establishing an unique twice coagulation (TC) strategy through the use of transformable gel precursors. This endeavor is conceptually new, which points out a new direction for aerogel fabrication and should greatly advance the controlled fabrication of aerogels with tailored properties/applications. Also, the manuscript is well organized and the conclusion is well supported by the experiments/analysis. After minor revision as shown as follows, I recommend the publication of this paper in Nature Communications.

Response: We appreciate the reviewer's positive evaluation of our manuscript. We thank the reviewer very much for the constructive comments and suggestions to improve our manuscript greatly. We have addressed all the issues in a point-by-point manner and all the changes to the original manuscript are highlighted in blue.

Question 1. The detailed information of TGA and Raman scattering spectra should be provided in Characterization part.

Response: Thank you for raising this point. The detailed information of TGA and

Raman scattering spectra have been provided in Characterization part.

Revision: “Thermal stability was determined by the thermal gravimetric analyzer (TGA, 209F1, NETZSCH, Germany) with the heating rate of 10 K min^{-1} in a nitrogen atmosphere.

Raman spectra were recorded on a LabRam HR Raman spectrometer with 50 W He-Ne laser operating at 632 nm with a CCD deter.” (Line 18, Page 25)

Question 2. Schematic diagram was recommended to intuitively represent the difference between cellular network and 3D fiber network.

Response: Thank you for your suggestion. The schematic diagram has been provided as follow and in **Supplementary Scheme 2** in the revised manuscript.

Revision:

Supplementary Scheme 2. Schematic diagram of the difference between 3D fiber network and cellular network.

Question 3. It is better to provide evidence to prove the successful fabrication of Paraffin@AP.

Response: Thank you for your suggestion. To prove the successful fabrication of Paraffin@AP, SEM image is provided and added in the Revised Supplementary Information.

Revision: The successful fabrication of Paraffin@AP was confirmed by SEM, demonstrating the infiltration of paraffin into the pores of the cellular aerogel network (Supplementary Fig. 30). (Line 22, Page 20)

Supplementary Fig. 30. SEM image of Paraffin@AP.

Question 4. Some format errors should be carefully checked, such as spaces between numbers and units.

Response: Thank you, the article has been rechecked carefully and the format errors has been modified.

REVIEWERS' COMMENTS

Reviewer #1 (Remarks to the Author):

The authors revised the paper nicely according to the reviewer comments. It was a pleasant reading from my side.

Reviewer #2 (Remarks to the Author):

According to the comments from three reviewers, the authors have revised the manuscript carefully by adding additional experiments and more analyses and discussions. The authors have addressed nearly all the concerns from three reviewers. The current study would have significant impact on the design and fabrication of novel aerogels with excellent mechanical properties and remarkable configuration-editable capacity. Therefore, the current manuscript can be recommended for publication.

Reviewer #3 (Remarks to the Author):

The authors have made significant efforts in revision and the quality of the manuscript has been improved substantially. It reaches the publication standard in Nature Communications currently.

The only tip from me is, the colorful shadows in the Figure 2d & 2f are not needed.

Reply to reviewers

To Reviewer #1:

Comments: The authors revised the paper nicely according to the reviewer comments. It was a pleasant reading from my side.

Reply: We really appreciate your positive comments on our revised manuscript and recommendation for its publication in *Nature Communications*.

To Reviewer #2

Comments: According to the comments from three reviewers, the authors have revised the manuscript carefully by adding additional experiments and more analyses and discussions. The authors have addressed nearly all the concerns from three reviewers. The current study would have significant impact on the design and fabrication of novel aerogels with excellent mechanical properties and remarkable configuration-editable capacity. Therefore, the current manuscript can be recommended for publication.

Reply: We really appreciate your positive comments on our revised manuscript and recommendation for its publication in *Nature Communications*.

To Reviewer #3:

Comments: The authors have made significant efforts in revision and the quality of the manuscript has been improved substantially. It reaches the publication standard in Nature Communications currently.

The only tip from me is, the colorful shadows in the Figure 2d & 2f are not needed.

Reply: We really appreciate your positive comments on our revised manuscript and recommendation for its publication in *Nature Communications*. We have removed the colorful shadows in the Figure 2d & 2f as shown in revised Figure 3d & 3f.